# Extended low-resolution structure of a *Leptospira* antigen offers high bactericidal antibody accessibility amenable to vaccine design

Ching-Lin Hsieh[1†], Christopher P Ptak[1,2†], Andrew Tseng[1], Igor Massahiro de Souza Suguiura[1], Sean P McDonough[3], Tepyuda Sritrakul[1], Ting Li[1], Yi-Pin Lin[4], Richard E Gillilan[5], Robert E Oswald[2]*, Yung-Fu Chang[1]*

[1]Department of Population Medicine and Diagnostic Sciences, College of Veterinary Medicine, Cornell University, Ithaca, United States; [2]Department of Molecular Medicine, College of Veterinary Medicine, Cornell University, Ithaca, United States; [3]Department of Biomedical Sciences, College of Veterinary Medicine, Cornell University, Ithaca, United States; [4]Division of Infectious Disease, Wadsworth Center, New York State Department of Health, Albany, United States; [5]Macromolecular Diffraction Facility at CHESS (MacCHESS), Cornell University, Ithaca, United States

*For correspondence:
reo1@cornell.edu (REO);
yc42@cornell.edu (Y-FC)

†These authors contributed equally to this work

Competing interests: The authors declare that no competing interests exist.

**Abstract** Pathogens rely on proteins embedded on their surface to perform tasks essential for host infection. These obligatory structures exposed to the host immune system provide important targets for rational vaccine design. Here, we use a systematically designed series of multi-domain constructs in combination with small angle X-ray scattering (SAXS) to determine the structure of the main immunoreactive region from a major antigen from *Leptospira interrogans*, LigB. An anti-LigB monoclonal antibody library exhibits cell binding and bactericidal activity with extensive domain coverage complementing the elongated architecture observed in the SAXS structure. Combining antigenic motifs in a single-domain chimeric immunoglobulin-like fold generated a vaccine that greatly enhances leptospiral protection over vaccination with single parent domains. Our study demonstrates how understanding an antigen's structure and antibody accessible surfaces can guide the design and engineering of improved recombinant antigen-based vaccines.
DOI: https://doi.org/10.7554/eLife.30051.001

## Introduction

The molecular details of how surface antigens of a pathogen are exposed to the host's defenses are highly relevant to rational vaccine development (*Rappuoli et al., 2016*; *Dormitzer et al., 2012*; *Dormitzer et al., 2008*). The structural context of immunologically accessible epitopes allows for the redesign of recombinant vaccine scaffolds to display the most antigenic surfaces (*Correia et al., 2014*; *Malito et al., 2014*). The exploration of recombinant strategies can be particularly fruitful when classical vaccine strategies provide weak protection. Currently available inactivated vaccines against leptospirosis, the most common bacterial zoonosis (*Picardeau, 2017*), are inadequate because of the severe side-effects and lack of cross-protection among pathogenic *Leptospira* species (*Grassmann et al., 2017*; *Adler, 2015*). Because the estimated worldwide burden of leptospirosis is over 1 million severe cases and ~60,000 deaths per year, advances in recombinant leptospiral vaccines are desperately needed and are likely to benefit from a structure-based antigen design strategy (*Picardeau, 2017*; *Costa et al., 2015*).

**eLife digest** Vaccines encourage the immune system to develop a protection against disease-causing bacteria and viruses. Some types of immune cells release antibodies, which recognize particular proteins on the surface of the invading microbe. A vaccine that contains these surface proteins allows immune cells to develop the antibodies that can help to fight off an infection at a later date. Studying the shape and structure of the surface proteins can reveal how they are detected by our immune systems and can further be used to design more effective vaccines.

Leptospirosis is the most common bacterial disease to affect both humans and animals. The symptoms of the disease include fever, muscle pain and bleeding from the lungs. New vaccines against leptospirosis are desperately needed because current ones have severe side effects and do not fully protect against the disease. The most promising new vaccine candidates are the Lig proteins, which are found on the surface of leptospirosis causing bacteria cells, but little was known about their molecular structure.

The region of the Lig protein that is recognized by the immune system consists of a series of twelve connected 'immunoglobulin-like' domains. Hsieh, Ptak et al. used X-ray scattering to determine the structure of this region and found that the protein is highly elongated. Additional experiments showed that the individual domains provoke immune responses to different extents. Antibodies that can interact more strongly with the Lig protein were also better able to kill the bacteria. Based on this information, Hsieh, Ptak et al. combined parts of the individual domains that bind strongly to antibodies to design a new protein that, when used as a vaccine, protected hamsters against leptospirosis much better than other Lig protein-based vaccines.

Further engineering and testing are required to develop an optimized, commercial leptospirosis vaccine, but the work of Hsieh, Ptak et al. shows the effectiveness of structure-based vaccine design methods. In the future, similar methods could be used to develop better vaccines and treatments for other infectious diseases.

DOI: https://doi.org/10.7554/eLife.30051.002

While several leptospiral antigens have been explored for use in vaccines, the most promising candidates have been derived from the *Leptospira* immunoglobulin-like (Lig) protein family (*Ko et al., 2009*; *Grassmann et al., 2017*; *Cao et al., 2011*; *Yan et al., 2009*; *Chang et al., 2007*). Lig proteins are present in only pathogenic species with LigB (but not LigA or LigC) being found in all pathogenic *Leptospira* genomes (*McBride et al., 2009*; *Matsunaga et al., 2003*), LigB's expression during host invasion further suggests an important role in virulence (*Lessa-Aquino et al., 2017*; *Choy et al., 2007*). To promote infection, LigB can bind multiple blood factors and extracellular matrix molecules (ECMs) to facilitate immune system evasion and tissue colonization (*Choy, 2012*; *Vieira et al., 2014*; *Figueira et al., 2011*). The persistence and exposure of LigB on the leptospiral outer membrane during infection leaves an exploitable vulnerability. Recently, a hamster immunization study has suggested that LigB-derived vaccines have the potential to confer sterile immunity against leptospiral challenge (*Conrad et al., 2017*). The new study (*Conrad et al., 2017*) is inconsistent with earlier LigB vaccine studies which only confer partial protection (*Yan et al., 2009*; *Silva et al., 2007*) and raises the possibility of further improvements in LigB-derived vaccine efficiency through rational design.

Our understanding of the LigB structure is limited to the NMR structure of the individual Lig protein Ig-like domain (*Ptak et al., 2014*). LigB is attached to the leptospiral outer surface by a short N-terminal anchor, which is followed by a stretch of twelve consecutive Ig-like domains, and flanked at its C-terminus by an additional non-Ig-like domain (*Figure 1—figure supplement 1A*). Most host interactions with Lig protein have been identified within the Ig-like domain regions and previously reported LigB-based vaccines have targeted the Ig-like domain region (*Yan et al., 2009*; *Conrad et al., 2017*; *Breda et al., 2015*; *Lin et al., 2009a*). A more comprehensive understanding of the domain arrangement would provide a picture of the most accessible antigenic regions and a guide for structure-based vaccine design.

In this study, small angle X-ray scattering (SAXS) (*Skou et al., 2014*) was used to obtain a low-resolution solution structure of the Ig-like domain region's architecture. The full LigB Ig-like domain

region's extended arrangement with notable bends encouraged the exploration of the highly-exposed surface for immunoreactivity with a library of anti-LigB monoclonal antibodies (mAbs). The capability of these mAbs to bind antigen and to adhere to the surface of pathogenic *Leptospira* was then correlated with their ability to kill these bacteria in the presence of serum complements. Finally, the identified mAb-reactive domains and previously obtained LigB12 (LigB 12th Ig-like domain) NMR structure informed the generation of chimeras on single Ig-like domain scaffolds capable of eliciting an immune response with either side of the β-sandwich domain. To illustrate the potential for rationally engineered antigens, a vaccine containing the chimera LigB10-B7-B7, which displays identified mAb-interacting surfaces from LigB7 and LigB10, offered greatly improved protection over LigB7 and LigB10 against leptospiral lethal challenge in hamsters. These findings provide a blueprint for combining immunoreactivity mapping from an mAb library and high-resolution structural information from NMR to engineer epitopes and improve the efficacy of LigB vaccines as well as recombinant vaccines from other pathogens.

## Results

### Extended structure of the LigB Ig-like domain region

To determine the architecture of the twelve Ig-like domain stretch in LigB, small angle X-ray scattering (SAXS) was used to generate low-resolution solution structures along a sliding multi-domain window. Because the 5-domain length contains four domain-domain linkers, the 5-domain structure proved to be optimal to define the relative angle of two neighboring domain-domain joints with one additional joint on each end (See Materials and methods for rationale; *Figure 1—figure supplement 1*).

All eight possible 5-domain protein constructs (LigB1-5 to LigB8-12) were analyzed with SAXS (*Figure 1*; *Figure 1—figure supplement 2*). Guinier fits and Porod analysis suggested minimal aggregation for all but LigB5-9 (*Figure 1—figure supplement 2B* and *Figure 1—source data 1*). The simulated fits to the experimental SAXS curves for the eight 5-domain constructs (*Figure 1A*; *Figure 1—figure supplement 2A*) were used to generate the atomic distance distribution for the molecules (*Figure 1B*). The longest atomic pair distance deduced from the pair-distance distribution function (*P(r)*) is indicative of the length of the 5-domain proteins. The expected length of a fully extended arrangement of five folded LigB Ig-like domains is only slightly longer than the pair distance for most of the eight 5-domain proteins and their corresponding envelopes (*Figure 1B*; *Figure 1—figure supplement 2C*). LigB5-9 has an atomic pair distance that exceeds what is possible for a folded 5-domain monomer (in agreement with aggregation indicated by the difference between experimental and predicted molecular weights, *Figure 1—source data 1*) and was excluded from the final model.

Ambiguity assessment using the AMBIMETER program (*Petoukhov and Svergun, 2015*) shows that the first six structures (LigB1-5 to LigB6-10) have a high degree of uniqueness with ambiguity scores of 0.3 or lower. The last two structures (LigB7-11 and LigB8-12) have scores of 1.7 and 1.4 respectively, which indicates the potential for shape ambiguity. Each of the eight 5-domain SAXS envelopes exhibited distinct structures with a variety of domain-domain angles (*Figure 1—figure supplement 2D*). A significant degree of bending is present between the first three domains. The stretch of domains between LigB3-6 is particularly straight. Slight bending in the angle between domains in the final five domain stretch produces a gentle spiral shape. By aligning the bends in the four shared domains of neighboring 5-domain regions, a best fit structure was generated for the full twelve Ig-like domains of LigB (*Figure 1C*). Overall the LigB1-12 structure is only 16% shorter than a fully extended model structure of twelve folded domains. The SAXS-derived structure of LigB1-12 demonstrates that most of the individual Ig-like domains exhibit a high degree of exposed surface area and suggests a high degree of accessibility to host interactions. The extensive exposed surface area is a consequence of the unexpected rigid, rod-like structure, with several well-defined kinks.

### Creation of mAb libraries against LigB

To explore the exposure of the LigB Ig-like domain region to a host immune response, two purified LigB truncations, LigB1-7 and LigB7-12 (*Figure 2A*), were used to generate two sets of hybridoma cell lines for mAb production: library C (24 mAbs) and library V (36 mAbs), respectively. Using ELISA,

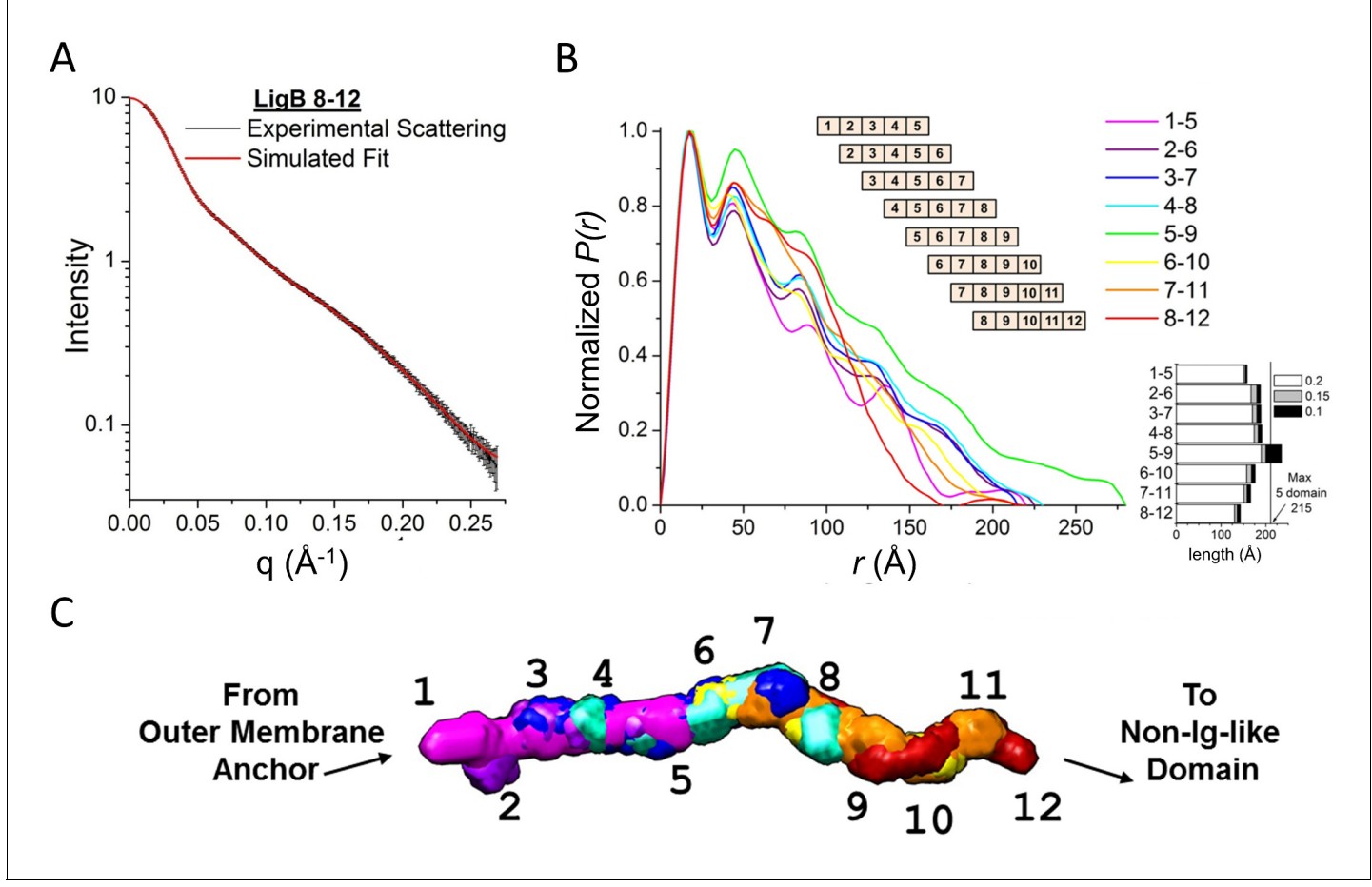

**Figure 1.** Full LigB Ig-like domain region (LigB1-12) determined from experimental SAXS data of 5-domain constructs. (**A**) The experimental scattering data for LigB8-12 (black) is shown with the simulated fit (red). The scattering curve is an average of 15 scans ± S.D. (**B**) The pair distance distributions, $P(r)$, were calculated from SAXS plots of all possible 5-domain LigB Ig-like domains using GNOM. A representation of the 5-domain constructs is shown in the inset. An estimate of construct length based on the longest atomic distances at 20%, 15%, and 10% max population height are shown. (**C**) DAMFILT envelopes were combined to create a representative envelope of the twelve Ig-like domains. Construct LigB5-9 was not included because the maximum distance distribution exceeds the expected length of a 5-domain construct.

DOI: https://doi.org/10.7554/eLife.30051.003

The following source data and figure supplements are available for figure 1:

**Source data 1.** LigB five domain construct SAXS profile values.

DOI: https://doi.org/10.7554/eLife.30051.006

**Figure supplement 1.** SAXS construct design.

DOI: https://doi.org/10.7554/eLife.30051.004

**Figure supplement 2.** Experimental SAXS data and structures for all LigB Ig-like domain 5-domain constructs.

DOI: https://doi.org/10.7554/eLife.30051.005

hybridoma supernatants containing anti-LigB mAbs were qualitatively screened for the ability to bind to their respective LigB truncations (***Figure 2B***). Based on the distribution of antigen binding efficiencies, a threshold level was set to $OD_{630} = 1.0$ (***Figure 2—figure supplement 1***). Only mAbs with binding above the threshold level were purified for further characterization of binding properties and bactericidal activity. Screening of library V required an additional twelve mAbs in order to equal the nine threshold-level mAbs identified in the library C screen.

## Binding properties of anti-LigB mAb libraries

For each library, nine mAbs were identified to have moderate to high binding efficiency to LigB antigens during the initial mAb screen. The dissociation constants ($K_D$) for each of these mAbs was

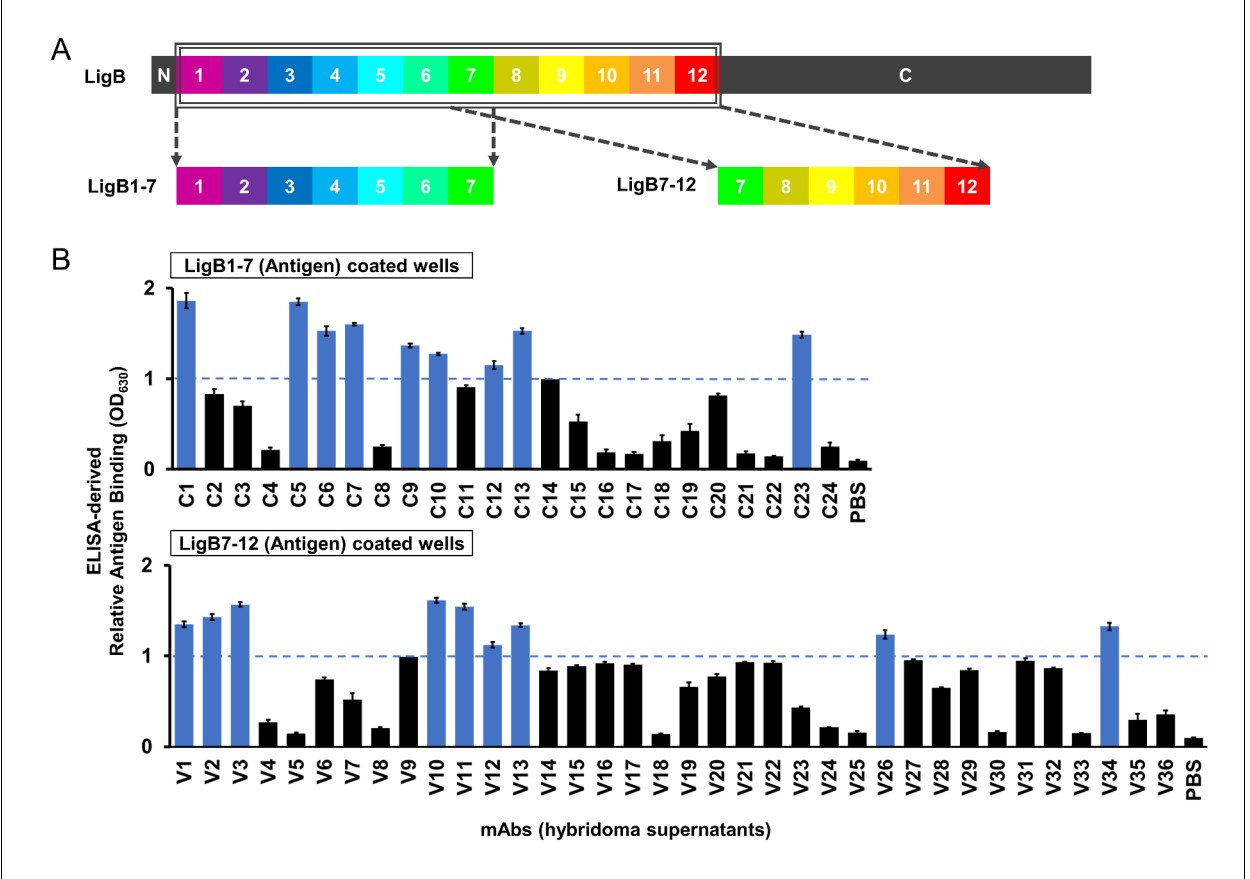

**Figure 2.** ELISA screen of anti-LigB mAb library. (**A**) Depiction of the full length LigB antigen and LigB-derived antigens used for mAb generation. (**B**) The degree of mAb binding to the respective LigB antigens was indirectly measured by colorimetic ($\lambda_{630}$) TMB-ELISA using rabbit anti-mouse IgG antibody conjugated with HRP (1:5000). Each value represents the mean ±S.D. from three individual trials of two replicates. Additional mAb characterization was limited to mAbs with binding efficiencies resulting in $OD_{630}$ >1.0 (blue).

DOI: https://doi.org/10.7554/eLife.30051.007

The following figure supplement is available for figure 2:

**Figure supplement 1.** Histogram of relative antigen binding strength for the anti-LigB mAb libraries.

DOI: https://doi.org/10.7554/eLife.30051.008

obtained from dose-dependent ELISA curves (*Figure 3*; *Figure 3—figure supplement 1*; *Table 1*). Based on $K_D$ values, library C mAbs were generally able to bind tighter to the LigB antigen than library V mAbs. Several library C mAbs (C5, C6, C7) exhibit sub-micromolar $K_D$ values while only one library V mAb (V10) was able to bind in the sub-micromolar range.

Domain-level specificity of individual LigB mAbs was investigated using a comprehensive set of single Ig-like domain LigB truncates and tandem (double) Ig-like domain LigB truncates (except LigB5-6) based on the NMR structure (*Ptak et al., 2014*) of the single domain, LigB12 (*Figure 4—figure supplement 1*). The LigB-derived antigens were immobilized on microtiter plates and tested for mAb binding specificity using an ELISA binding assay. The ELISA results for individual mAbs were generally able to localize binding to a specific double and/or single Ig-like domain. *Figure 4A* illustrates the ELISA assays used to identify the binding of mAbs C5 and V10 for specific two-domain regions and for specific single Ig-like domains. A comprehensive survey of domain-level epitope mapping for different mAbs are summarized in *Table 1* and *Figure 4—figure supplement 1*. For the LigB1-7 antigen-derived library, immunoreactivity of mAbs was weighted towards LigB1-2 and LigB4-5 regions, while for the LigB7-12 antigen-derived library, mAbs were preferentially generated against LigB7-8 and LigB10-11 regions. Only 3 of 10 double domains, LigB3-4, LigB6-7, and LigB8-9,

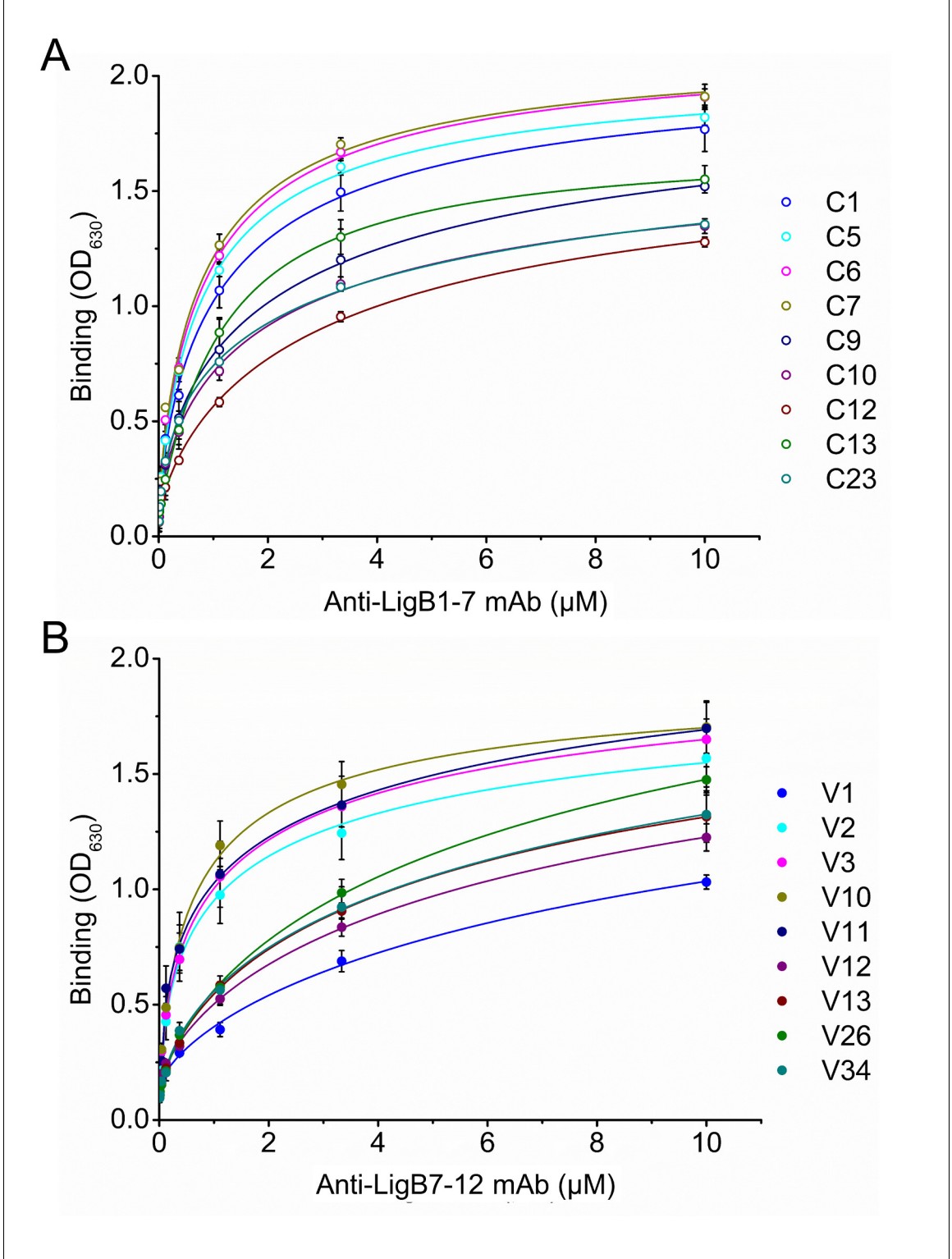

**Figure 3.** Equilibrium binding for anti-LigB mAbs. The equilibrium dissociation constants ($K_D$) for mAbs from library C (**A**) and library V (**B**) were determined from dose-dependent binding curves. Increasing concentrations of purified anti-LigB mAbs (0.00686, 0.0137, 0.0412, 0.123, 0.370, 1.11, 3.33 and 10 μM) were incubated with LigB antigen (1 μM) immobilized on microtiter plates. The binding interaction was subsequently detected by ELISA using HRP-conjugated anti-mouse antibodies. All experiments were conducted in three trials, the mean ±S.D. of which were shown in bar charts.
DOI: https://doi.org/10.7554/eLife.30051.009

The following figure supplement is available for figure 3:

*Figure 3 continued on next page*

*Figure 3 continued*

**Figure supplement 1.** $K_D$ values of anti-LigB mAbs.

DOI: https://doi.org/10.7554/eLife.30051.010

and 2 of 12 single domains, LigB3 and LigB11, lack immunogenicity for the set of tested mAbs. From an antigenic response to LigB1-7 and LigB7-12, at least every other individual Ig-like domain is capable of eliciting mAb production and the generated-LigB mAb libraries cover the length of the

**Table 1.** Anti-LigB mAb characterization summary.

Mean values for the dissociation constant ($K_D$), FACS cell binding propensity (MFI), and lethal dose ($LD_{50}$) are listed for mAbs against LigB1-7 (library C) and LigB7-12 (library V). Double and single domain specificities for mAbs are also noted.

| Anti-LigB mAbs | $K_D$ (μM) | Cell binding (MFI) | $LD_{50}$ (μg/ml) | LigB domain specificity | |
|---|---|---|---|---|---|
| **Library C (Anti-LigB1-7)** | | | | **Double domain** | **Single domain** |
| C1 | 1.232 | 3497.5 | 15.29 | LigB1-2 LigB2-3 | LigB2 |
| C5 | 0.896 | 4499.5 | 15.07 | LigB4-5 | LigB5 |
| C6 | 0.923 | 4644.5 | 13.76 | LigB4-5 | LigB5 |
| C7 | 0.848 | 2722.5 | 18.72 | LigB1-2 | LigB1 |
| C9 | 2.440 | 3530.0 | 19.56 | LigB1-2 | n.s. |
| C10 | 2.162 | | 21.63 | LigB4-5 | n.s. |
| C12 | 3.301 | | 11.57 | LigB2-3 LigB4-5 | LigB4 LigB6 |
| C13 | 1.166 | 4186.0 | 18.98 | LigB1-2 LigB4-5 | n.s. |
| C23 | 2.916 | 3469.5 | 29.92 | LigB2-3 | n.s. |
| C22 | n.b. | 1121.5 | | | |
| Library V (Anti-LigB7-12) | | | | Double Domain | Single Domain |
| V1 | 15.151 | | 25.85 | LigB9-10 LigB10-11 | LigB10 |
| V2 | 1.391 | 2889.0 | 20.45 | LigB11-12 | LigB12 |
| V3 | 1.367 | 2568.5 | 16.19 | LigB7-8 LigB11-12 | LigB7 |
| V10 | 0.738 | 3211.0 | 14.87 | LigB9-10 LigB10-11 | LigB10 |
| V11 | 2.956 | | 19.24 | LigB7-8 LigB11-12 | LigB8 LigB12 |
| V12 | 9.574 | 2333.5 | 29.18 | LigB7-8 | LigB7 LigB9 |
| V13 | 7.064 | 2376.0 | 28.60 | LigB7-8 | LigB8 |
| V26 | 9.915 | | 22.83 | LigB9-10 LigB10-11 | n.s. |
| V34 | 11.011 | 2665.0 | 25.28 | LigB7-8 | LigB7 LigB9 |
| V33 | n.b. | 910.5 | | | |
| Source of relevant data | *Figure 3, Figure 3—figure supplement 1* | *Figure 4B, Figure 4—figure supplement 2* | *Figure 5, Figure 5—figure supplement 1* | *Figure 4A, Figure 4—figure supplement 1* | |

n.b. no significant binding.

n.s. no significant binding partner was determined.

DOI: https://doi.org/10.7554/eLife.30051.011

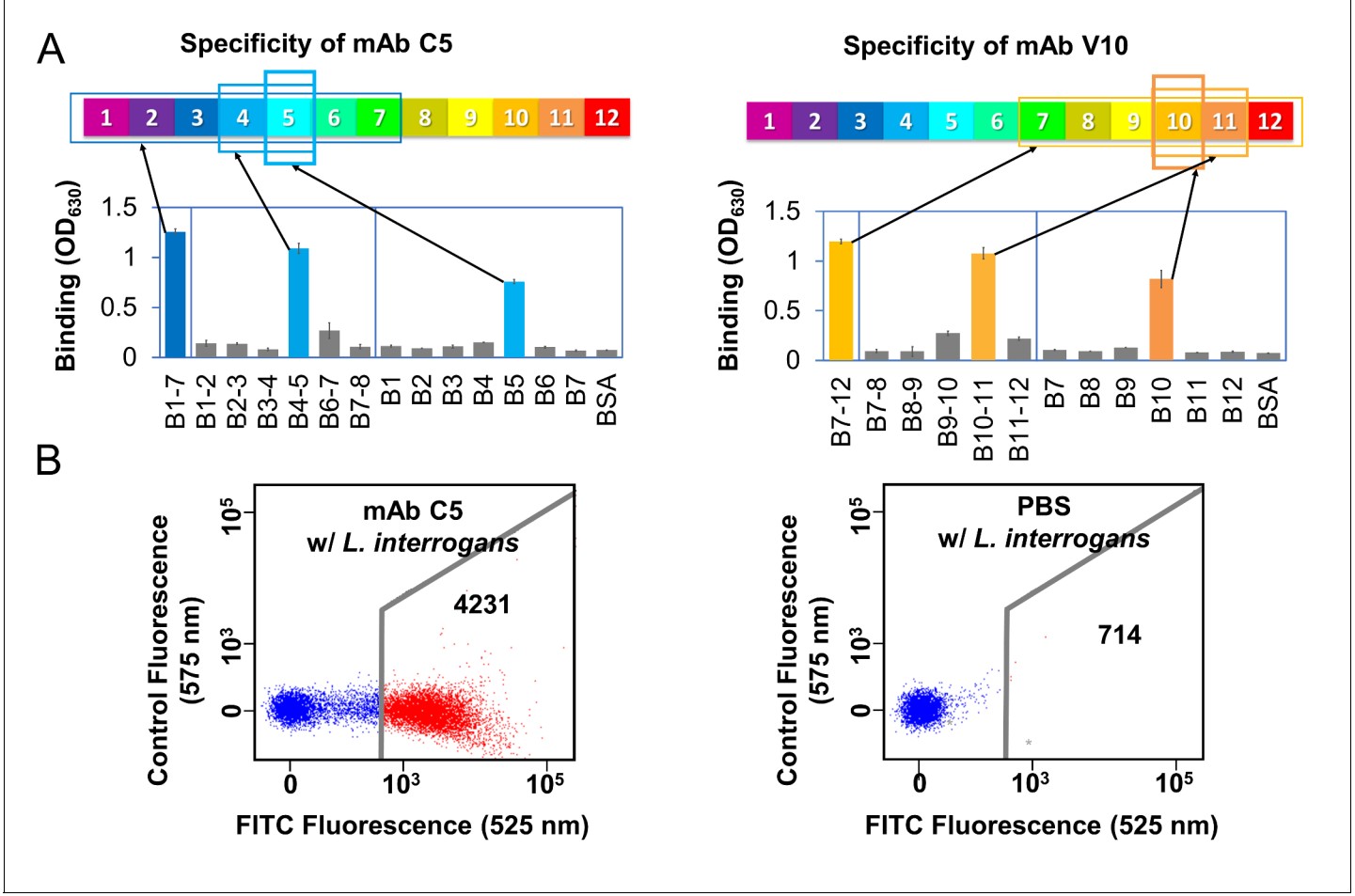

**Figure 4.** Anti-LigB mAb binding characterization. (**A**) A representative example of domain-level epitope ELISA-based mapping for anti-LigB mAbs (full dataset *Figure 4—figure supplement 1*). The mean ±S.D. for three trials is displayed. (**B**) Flow cytometry was used to measure the ability of anti-LigB mAbs to bind directly to live *Leptospira* (*Figure 4—figure supplement 2B*). The representative fluorescence emission displays the relative binding of fluorescently-labelled secondary anti-mouse IgG antibodies to PBS-treated cells (negative control, right panel) and to mAb C5-treated cells (left panel).
DOI: https://doi.org/10.7554/eLife.30051.012

The following figure supplements are available for figure 4:

**Figure supplement 1.** Domain-level epitope mapping of anti-LigB mAb library determined by ELISA.
DOI: https://doi.org/10.7554/eLife.30051.013

**Figure supplement 2.** Cell surface binding of anti-LigB mAbs.
DOI: https://doi.org/10.7554/eLife.30051.014

LigB Ig-like domain region. Several anti-LigB mAbs were tested using fluorescence-based flow cytometry for the ability to recognize native proteins on the surface of *Leptospira* cells. The spirochetes were incubated with mAbs from library C or V. Anti-LigB mAb-bound *Leptospira* cells were fluorescently-labelled with anti-mouse IgG antibodies and counted by flow cytometry. The strong fluorescence signal from incubation of the pathogenic *L. interrogans* serovar Pomona cells with anti-LigB mAb C5 was indicative of a tight cell surface interaction (*Figure 4B*). Cells incubated with either PBS (*Figure 4B*) or the negative control mAb C22 (*Figure 4—figure supplement 2A*) failed to generate a fluorescence signal after secondary anti-mouse IgG antibody incubation. Additionally, cells from *L. biflexa*, a non-infectious *Leptospira* species which lacks the genes for Lig proteins (*Figueira et al., 2011*), also failed to exhibit a fluorescence signal strong enough to indicate binding after anti-LigB mAb C5 incubation (*Figure 4—figure supplement 2A*). A total of seven library C mAbs and six library V mAbs were measured for *Leptospira* surface binding character and all produced a fluorescence signal count over three-fold higher than the PBS control and over two-fold

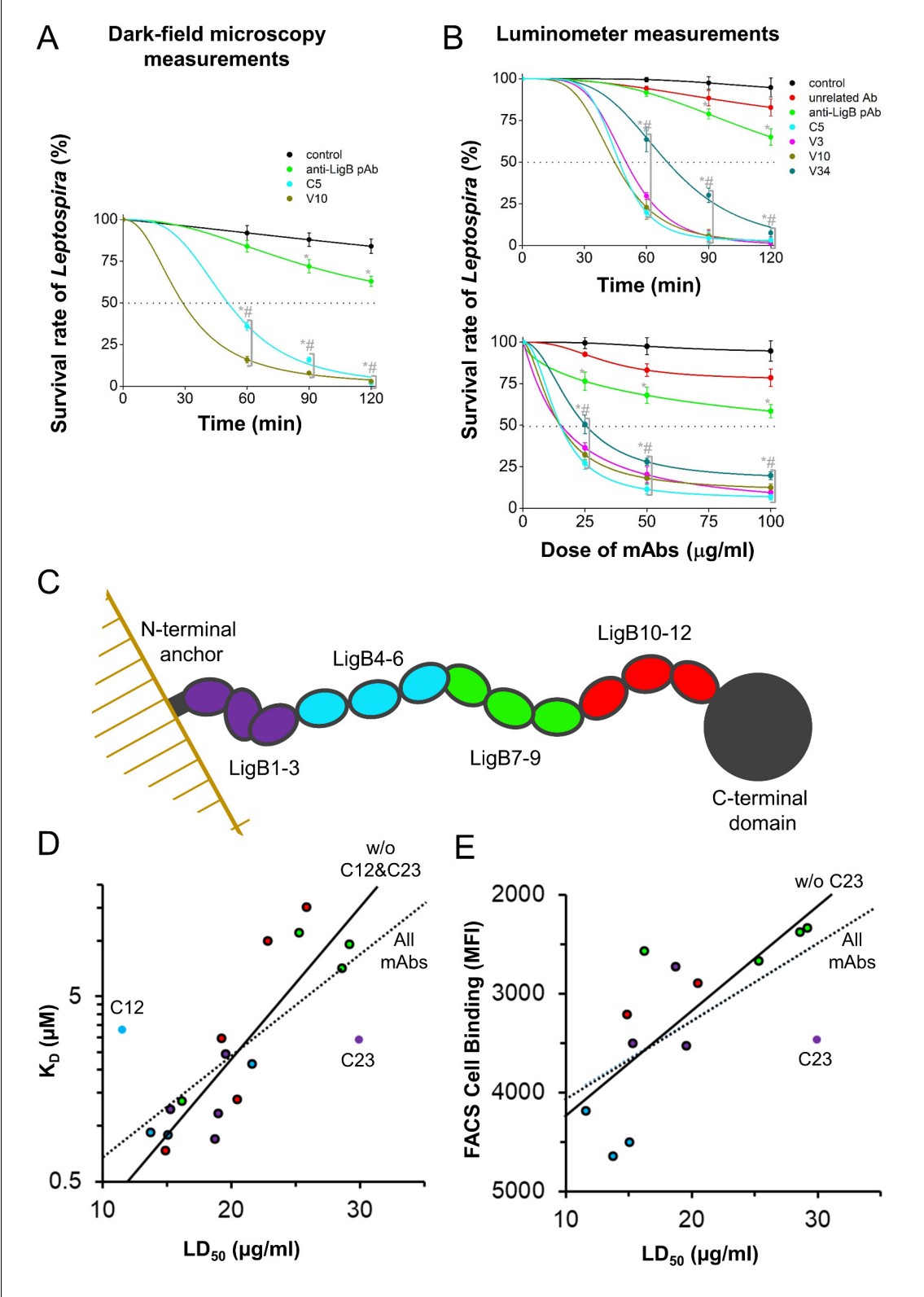

**Figure 5.** Anti-LigB mAb bactericidal properties. (A, B) *Leptospira* survival assays were used to assess mAb bactericidal activity. (A) After incubation with individual mAbs, the survival rate of *L. interrogans* Pomona was observed using dark-field microscopy (B) while the survival rate of bioluminance-producing *L. interrogans* Manilae was measured using a 96-well luminometer-based assay (full dataset *Figure 5—figure supplement 1*). Each value represents the mean ± S.D. from three individual trials of two replicates. Statistically significant (*t*-test; $p < 0.05$) differences were calculated from the

*Figure 5 continued on next page*

*Figure 5 continued*

comparisons between mAb-treated groups and the control group (*) or between mAb-treated groups and the pAb group (#). (**C**) A schematic is shown for the SAXS-derived LigB structure. (**D**) For the set of mAbs, the ELISA antigen binding value is plotted against the $LD_{50}$. Points for mAbs are colored to match the three domain regions defined in the schematic. Statistically-determined outliers are labelled and data was fit with and without outliers (dashed line, $R^2$=0.508 and solid line, $R^2$=0.773, respectively). (**E**) For mAbs measured for cell surface binding, the FACS cell binding value is plotted against the $LD_{50}$ and data is fit with and without outlier, mAb C23 (dashed line, $R^2$=0.312 and solid line, $R^2$=0.466, respectively).

DOI: https://doi.org/10.7554/eLife.30051.015

The following figure supplements are available for figure 5:

**Figure supplement 1.** Serum bactericidal activity characterization of anti-LigB mAb library.

DOI: https://doi.org/10.7554/eLife.30051.016

**Figure supplement 2.** Statistically determined outliers for scatter plots.

DOI: https://doi.org/10.7554/eLife.30051.017

**Figure supplement 3.** Binding of library C mAbs to LigB7-12.

DOI: https://doi.org/10.7554/eLife.30051.018

**Figure supplement 4.** Comparison of bactericidal activity values, $LT_{50}$ and $LD_{50}$.

DOI: https://doi.org/10.7554/eLife.30051.019

higher than the poorly binding mAb controls (C22 and V33) (*Figure 4—figure supplement 2B*). The cell binding propensities for measured mAbs are listed as mean fluorescence intensity (MFI) values in *Table 1*. The flow cytometry data supports the presence of mAb-accessible Lig protein Ig-like domains on the surface of *Leptospira* cells.

## Bactericidal activity of anti-LigB mAbs

To test the role of the mAb libraries in promoting complement-mediated killing of *Leptospira in vitro*, the bactericidal activity of the LigB-binding mAbs was tested by incubating live *L. interrogans* serovar Pomona cells with complement-containing human serum with or without added antibodies. Using dark-field microscopy, motile bacteria were counted to calculate the survival rate at different time points. The PBS only (negative control) condition was unable to effectively kill the live *Leptospira* cells leaving an 84% survival rate at 120 min post treatment (*Figure 5A*). The hamster-derived polyclonal antibody (pAb) treatment of *Leptospira* cells led to an enhancement of bactericidal activity with only 63% survival at 120 min post treatment. *Leptospira* incubated with C5 or V10 mAbs showed only a 2% to 3% survival rate (a >20 fold improvement over pAb-incubated spirochetes p<0.05) (*Figure 5A*).

In addition, a high-throughput luciferase-based methodology was utilized to screen the bactericidal activity of the full set of mAbs. The assay measures the luminescence intensity from the expression of the light-generating *lux* cassette proteins in another pathogenic *L. interrogans* serovar Manilae to accurately report the viable cell count (*Murray et al., 2010*). Note that the LigB proteins from *L. interrogans* serovar Manilae and *L. interrogans* serovar Pomona share 96% identity suggesting that the bactericidal properties provided by the LigB (Pomona) mAb library using these two serovars would be similar. Bioluminescent *L. interrogans* serovar Manilae were again incubated with complement-containing human serum with or without added antibodies. The survival rate was determined by measuring the luminescence intensity of metabolically active *Leptospira* relative to that from the spirochetes prior to Ab-incubation. Both incubation time and antibody concentration were varied to obtain time-dependent and dose-dependent curves (*Figure 5—figure supplement 1A and C*, respectively) as well as $LT_{50}$ and $LD_{50}$ values (*Table 1*). Experimental analysis of luminometer measurements for control conditions and representative mAbs are presented in *Figure 5B*. The two negative controls, PBS only and non-specific mouse IgG1, as well as the hamster-derived pAbs were unable to kill >50% of cells at the longest time point (120 min) or at the highest dosage (100 μg/ml). All anti-LigB mAbs were able to eliminate >50% of *Leptospira* between 40 and 85 min with the exception of mAb C23 (*Figure 5—figure supplement 1B*). Each of the LigB-specific mAbs had the potential to kill >90% of the active *Leptospira* cells after a sufficient exposure time. All of the LigB-specific mAbs were able to effectively decrease the number of metabolically active *Leptospira* cells to less than 50% with a dose value ($LD_{50}$) between 10 and 30 μg/ml (*Figure 5—figure supplement 1D*). The relative lack of efficiency of hamster pAbs compared to purified mAbs could be explained by the pAbs being a mixture of high affinity and low affinity antibodies, being a mixture of IgG

subclasses with differing abilities to activate the complement system, or being derived from different rodents than the mAbs with potentially different abilities to activate the complement system. The ability of the anti-LigB mAbs to kill *Leptospira* in the presence of complement is consistent with their capability in binding to LigB protein in pathogenic *Leptospira*.

## Activity-Interaction map analysis of anti-LigB mAbs

In the context of a known antigen structure, the results of mAb-antigen interaction and mAb bactericidal activity experiments provide the opportunity to explore potential correlations between these three data sets. The broad mAb accessibility of LigB Ig-like domains supports the extended organization of the LigB1-12 SAXS structure. The LigB structure's domain-domain arrangement is summarized in 3-domain segments (*Figure 5C*). The properties for each mAb have been correlated in plots of $K_D$ vs. $LD_{50}$ (*Figure 5D*) or FACS cell binding vs. $LD_{50}$ (*Figure 5E*). Additionally, the mAb data points are colored to signify the mAb's LigB segments binding specificity. A trendline for $K_D$ vs. $LD_{50}$ scatter plots using all data and excluding the two statistically-determined outliers (*Figure 5D*; *Figure 5—figure supplement 2*) yields correlations with $R^2=0.508$ and $R^2=0.773$, respectively. Antibodies specific to each of the four LigB segments are near the trendline. Thirteen anti-LigB mAbs were also tested for *Leptospira* surface binding ability. When plotted against $LD_{50}$, FACS-derived cell binding values fit to a trendline with an $R^2=0.312$ and the single outlier-adjusted trendline of with an $R^2=0.466$ (*Figure 5E*; *Figure 5—figure supplement 2*). In both plots, near-trendline mAbs with specificity to the LigB1-3 and LigB4-6 segments have a relatively high binding ability and low $LD_{50}$ and correlated mAbs with specificity to the LigB7-9 segment have a relatively low binding ability and high $LD_{50}$.

Because mAbs that do not fit the predicted correlation between binding affinity and bactericidal activity have the potential to inform vaccine studies, the bases for outlier discrepancies were examined further. Only one outlier, C12, displays enhanced bactericidal activity above what would be expected based from binding correlations. While C12 displays some specificity for the straightest segment (LigB4-6), it can bind to two single domains (LigB4 and LigB6) as well as a double domain in the segment LigB1-3. Further, an additional ELISA assay found that C12 is the only library C mAb that can also bind to LigB7-12 (*Figure 5—figure supplement 3*). The ability of C12 to bind to multiple Ig-like domains covering a large range of the Ig-like domain regions suggests a reason for the increased bactericidal activity relative to C12's binding affinity. The only outlier with bactericidal activity below expected is C23 which binds to LigB2-3. These domain neighbors exhibit the sharpest interdomain bend suggesting that steric limitations can differentially effect LigB binding and bactericidal activity. Indeed, the kinked structure of C23's target domain may decrease the rate for C23 to initiate bacterial killing relative to its $K_D$ (*Figure 5—figure supplement 4*). Overall, the link between antigen binding and bactericidal activity complements the highly accessible extended structure of LigB with highly accessible regions having the potential to yield high bactericidal mAbs. The high degree of correlation between antigen-mAb binding and bactericidal activity supports the hypothesis that a mAb's ability to kill a pathogen can be determined by its antigen-specific binding affinity.

## LigB Ig-like domain chimeras identify epitopes on each side of the 3-D fold

To provide proof that a single Ig-like domain can act as a scaffold to display multiple protective epitopes, the binding epitope for anti-LigB mAbs was more specifically mapped to a surface within individual Ig-like domains. Chimeric LigB Ig-like domains were designed to differentiate interactions specific to the surface residues of major folding units. The high degree of homology between LigB Ig-like domains (*Figure 6A and C*) provided an opportunity for tertiary structural elements to be exchanged at two chimeric swapping positions (*Ptak et al., 2014*) (*Figure 6A and B*). The first two segments of the chimeras (β-strands A-C and β-strands C′-F) were engineered to separate the top and bottom halves of the Ig-like domain β-sandwich. The third chimera region (β-strands G-G′) was included to determine if an antigenic surface is formed on the untethered edge of the β-sandwich. Four single domain proteins that have been recognized to contain mAb-specific epitopes (*Table 1*) were paired based on matching length to generate two sets of chimeras (LigB5/LigB12 and LigB7/LigB10). Including the wild-type domains, eight possible chimeric combinations were generated on single Ig-like domains (*Figure 6D and E*).

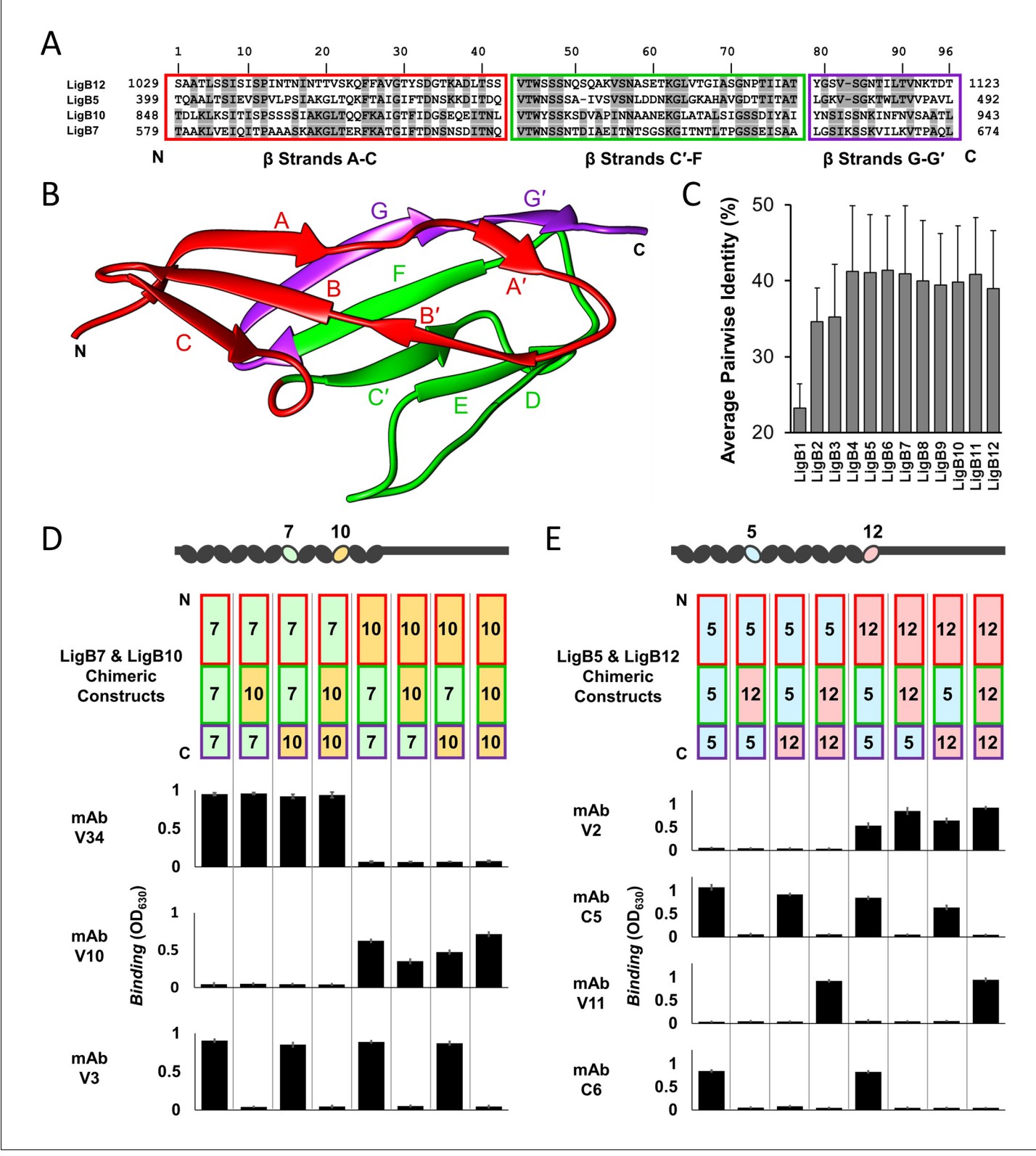

**Figure 6.** Dissection of mAb binding surfaces using chimeras. (**A**) Sequence alignment of LigB Ig-like domains used for generating chimeras. The chimeric swap regions are divided into colored boxes. Residues that are identical in either LigB5 and LigB12 or LigB7 and LigB10 pairs are shaded. (**B**) The structure of LigB12 (PDB ID 2MOG) is correspondingly colored by chimeric swap regions. (**C**) Average percent identity ± S.D. for each LigB Ig-like domain based on a pairwise matrix. (**D**) The set of LigB7 and LigB10 single domain chimeric constructs which cover all variations for the three swapped

*Figure 6 continued on next page*

*Figure 6 continued*

regions (schematic representation shown) were tested for binding to mAbs with LigB7-binding specificity (V34 and V3) or LigB10-binding specificity (V10) using an ELISA assay. (E) LigB5 and LigB12 chimeras were similarly tested for binding to mAbs with LigB5-binding specificity (C5 and C6) or LigB12-binding specificity (V2 and V11). The ELISA binding values correspond to the chimeric construct representation at the top of the column. All ELISA experiments were conducted in three trials, the mean ± S.D. of which were shown in bar charts.

DOI: https://doi.org/10.7554/eLife.30051.020

The following figure supplement is available for figure 6:

**Figure supplement 1.** Summary of mAb binding surfaces based on chimeras.

DOI: https://doi.org/10.7554/eLife.30051.021

All mAbs showed binding to a subset of the eight proteins with a clear region-specific pattern (*Figure 6D and E*). Two LigB7-specific mAbs (V12 and V34), two LigB10-specific mAbs (V1 and V10), and one LigB12-specific mAb (V2) could bind only to the N-terminal β-strands A-C from their respective interacting domains. The LigB5-specific mAb C5 and LigB7-specfic mAb V3 interacted with proteins that contained β-strands C′-F from LigB5 and LigB7, respectively. Unique to the LigB5/LigB12 mAb set, mAb V11 and mAb C6 required both the β-strands C′-F and the terminal β-strands G-G′ from LigB12 and LigB5, respectively. The mAb binding patterns demonstrate that the surface contribution of both β-strands A-C and β-strands C′-F (or C′-G′) can provide distinct epitopes for direct mAb targeting.

## Chimera LigB10-B7-B7 confers enhanced protection against *Leptospira* lethal challenge

The ultimate goal of combining structural and immunoreactivity studies of antigens is to explore a strategy for the rational engineering of improved vaccines. A chimeric LigB Ig-like domain was evaluated for the potential to elicit an immune response to multiple regions of LigB and to thereby enhance vaccine protection against leptospiral infection. The chimera LigB10-B7-B7 (β-strands A-C: B10 and β-strands C′-F and G-G′: B7) was chosen as the best candidate for animal studies because the chimeric domain possesses the ability to bind highly bactericidal mAbs specific to each parent domain (*Figure 6—figure supplement 1*). Additionally, LigB10-B7-B7 was similar to wild-type LigB7 and LigB10 in protein expression levels and in overall secondary structure (circular dichroism analysis, *Figure 7—figure supplement 1*). To generate a protective response, hamsters were immunized with 50 μg of LigB7, LigB10, LigB10-B7-B7, or PBS (as a negative control) for two times at 3 week intervals, and then challenged with $2.5 \times 10^2$ of triple passages of *L. interrogans* Pomona (*Conrad et al., 2017*; *Kunjantarachot, 2014*) (*Figure 7A*). On day 8, the leptospiral challenge was lethal for five of six hamsters inoculated with PBS group, three of five hamsters inoculated with wild-type LigB7, and four of five hamsters inoculated with wild-type LigB10. By day 10, all of the wild-type LigB7 and LigB10 immunized hamsters had either died or were euthanized due to severe clinical signs. In contrast, all five of the hamsters that had been inoculated with LigB10-B7-B7 and subsequently challenged with *Leptospira* survived until the end of the experiment (day 21 post infection). The survival rate of the LigB10-B7-B7 group is significantly higher (100%) than either control group (17%) or individual wild-type domain groups (0%) ($p<0.05$).

Serological fluids collected during the hamster studies were tested for the ability of the immunizations to generate an effective humoral response using a direct ELISA binding assay. The sera from LigB domain immunized hamsters provided a strong antibody response against the corresponding immobilized recombinant LigB domain (*Figure 7—figure supplement 2*). LigB7, LigB10, and LigB10-B7-B7 boosters were able to further enhance the antibody response. The Anti-LigB10-B7-B7 sera was also reactive with wild-type LigB7 and LigB10 but failed to react with the negative control, LigB12. LigB10-B7-B7 immunization led to an increase in post-booster secondary response for IgG but not IgM antibodies implying the generation of a typical immune memory response (*Figure 7—figure supplement 3*). The serological tests indicate that targeted antibodies can be effectively generated by vaccines based on individual LigB Ig-like domains and chimeric domains.

Several complementary methods were employed to test for the vaccine's ability to reduce bacterial burdens of host tissue. Liver, kidney, and urinary bladder tissue from all immunization groups were examined by real-time quantitative reverse transcription polymerase chain reaction (RT-qPCR) to identify the *Leptospira* specific gene, LipL32 (*Figure 7B*) (*Levett et al., 2005*; *Haake et al.,*

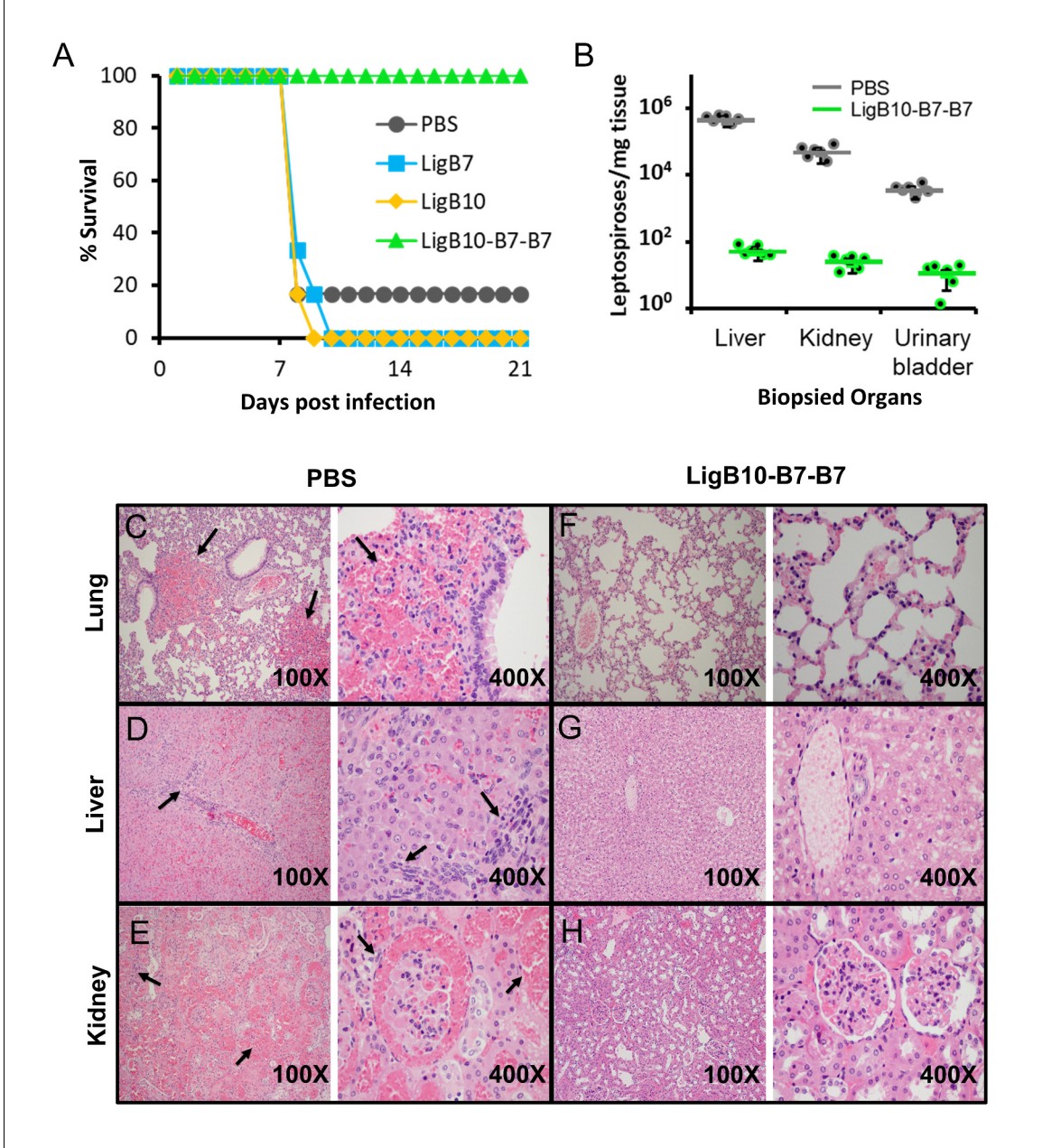

**Figure 7.** *Leptospira* lethal challenge of hamsters immunized with LigB Ig-like domains. (**A**) Survival rates for groups of six immunized hamsters are shown for the 3 weeks post leptospiral challenge. Inoculations and boosters with specific LigB Ig-like domains occurred 3 and 6 weeks prior to leptospiral infection. Sera from immunized hamsters were tested for domain-specific immunoreactivity (*Figure 7—figure supplement 2*). (**B**) The indicated organ tissues were biopsied from post-challenged hamsters immunized with PBS (control) or LigB10-B7-B7. The leptospiral load of hamster tissue was determined by RT-qPCR for the *Leptospira* specific gene, LipL32. Each point depicts the mean value obtained from duplicate analysis of individual tissue samples. Bars indicate the mean bacterial loads ± S.D. (six hamsters per group). Leptospiral loads for the PBS group are significantly higher than those of the LigB10-B7-B7 group in all three tissues. (**C-H**) Post-challenged hamster tissues that are targets for spirochete infection were fixed by formalin and stained by hematoxylin and eosin. PBS (control) immunized hamster tissue exhibited (**C**) multifocal lung hemorrhage, (**D**) liver inflammation and necrosis, and (**E**) tubulointerstitial nephritis and hemorrhage in the kidneys (each indicated by arrows). (**F-H**) Tissue from the LigB10-B7-B7 immunized group were all within normal limits.

DOI: https://doi.org/10.7554/eLife.30051.022

The following figure supplements are available for figure 7:

**Figure supplement 1.** Secondary structure analysis for LigB Ig-like domains utilized in immunization trials.

DOI: https://doi.org/10.7554/eLife.30051.023

*Figure 7 continued on next page*

*Figure 7 continued*

**Figure supplement 2.** Detection of humoral response.

DOI: https://doi.org/10.7554/eLife.30051.024

**Figure supplement 3.** Memory response triggered by LigB10-7-7 chimera.

DOI: https://doi.org/10.7554/eLife.30051.025

**Figure supplement 4.** Histograms of post-challenged hamster tissue scores for PBS, LigB7, LigB10, and LigB10-7-7 inoculated groups.

DOI: https://doi.org/10.7554/eLife.30051.026

*2000*). The hamster group that received immunizations containing recombinant chimeric protein, LigB10-B7-B7, exhibited a significant reduction in leptospiral burden when compared to bacterial loads from wild-type LigB7 or LigB10 immunized hamsters. The immunized hamsters were further examined for histopathologic changes of livers, kidneys and lungs (*Figure 7C–H* and *Figure 7—figure supplement 4*). Except for one apparently normal hamster from PBS group, all other hamsters from PBS, LigB7 and LigB10 immunization groups presented severe clinical signs and prominent macroscopic lesions on spirochete-targeted organs (e.g. multifocal pulmonary ecchymoses, icteric liver and enlarged kidney). Lung lesions included thickening of alveolar septa due to edema, interstitial leukocytes infiltration, endothelial cell swelling, and extensive hemorrhage (*Figure 7C*). Leptospira-infected livers were infiltrated by various inflammatory cells, indicating moderate to severe hepatitis. Focal necrosis was also found in parenchymal hepatocytes, leading to loss of normal tissue integrity (*Figure 7D*). Severe tubulointerstitial nephritis with locally extensive hemorrhage in uriniferous spaces and tubules was present in Leptospira-infected kidneys (*Figure 7E*). Furthermore, greater than 50% of renal tubules were lost and replaced by lymphoplasmacytic cells infiltration and severe fibrosis. In contrast, all hamsters immunized with chimeric LigB10-B7-B7 presented no visible macroscopic/microscopic lesions in livers, kidneys or lungs (*Figure 7F–H*). In agreement with high survival rate of LigB10-B7-B7 immunized hamsters, vaccines containing the chimeric antigen provide superior protection and potentially sterilizing immunity for the animals from leptospiral infection.

## Discussion

*Leptospira*, like other pathogenic bacteria, have evolved a wide variety of disease-related surface proteins to initiate colonization, to combat host defense systems and to reach target organs based on tissue-specific tropism (*Ko et al., 2009*; *Vieira et al., 2014*). Lig proteins contribute to pathogenesis by acting as both surface adhesins to mediate attachment to host extracellular matrix (ECM) molecules (*Lin et al., 2009a*; *Vieira et al., 2014*; *Figueira et al., 2011*) and recruiters of complement regulators to evade attack by the host's innate immune system (*Breda et al., 2015*; *Choy et al., 2007*; *Hsieh et al., 2016*; *Castiblanco-Valencia et al., 2012*). In addition, Ig-like domains from LigB are currently some of the strongest recombinant protein-based vaccine candidates in protecting hamsters from lethal challenges although discrepancies in effectiveness suggest that vaccine redesign efforts could help to create an improved vaccine (*Cao et al., 2011*; *Conrad et al., 2017*; *Yan et al., 2009*; *Adler, 2015*). This study provides new information regarding the architecture and accessibility of the main host-interacting region of LigB. The elongated structure presents insight into the mechanism of Lig protein adhesion, but, more importantly, an analysis of the mAb exposed regions and lack of long range inter-domain interactions has provided guidance for the optimization of recombinant leptospirosis vaccines using homologous Lig protein domains.

The domain arrangement of the LigB1-12 Ig-like domain region was determined using low resolution SAXS structures. The SAXS-derived LigB1-12 structure provides a visual depiction that complements the more therapeutically applicable information from mAb accessibility. The domain accessibility predicted by the LigB architecture also suggested that including the full set of Ig-like domains as antigens for mAb library generation was likely to be worthwhile. While previous SAXS studies have generated composite multi-domain structures (*Jeffries et al., 2011*; *Morgan et al., 2012*), the full LigB structure is the first example of using a sliding window of medium length, 5-domain constructs to generate a much longer 12-domain structure. The advantage of using a sliding window is that the high level of redundancy within the 5-domain structures allows for an increased accuracy in orienting and positioning the segments relative to each other. The sequence of domains LigB1-6 is identical to LigA domains 1–6 so the SAXS structure provides a partial structure of the

LigA Ig-like domain region. While LigA is not present in all pathogenic *Leptospira* species, LigA-based vaccines, like LigB-based vaccines, enhance immunity to leptospirosis (*Palaniappan et al., 2006*; *Silva et al., 2007*; *Faisal et al., 2009*).

Recombinant antigen-based vaccine technologies are the most promising avenue for the development of optimized protective strategies against leptospirosis. In recombinant antigen design, the basic antigen scaffold must present a native and well-folded epitope to the immune system while a minimized antigen can eliminate nonessential, disruptive features (*Kulp and Schief, 2013*). Examples of vaccines against different variants of *Neisseria meningitides* and Lyme disease *Borreliae* have been generated from chimeric recombinant antigens and have been capable of inducing broad spectrum bactericidal mAbs (*Earnhart and Marconi, 2007*; *Scarselli et al., 2011*). Ig-like domains are found in a variety of prokayotic and eukaryotic extracellular proteins and readily fold to their native structure (*Bodelón et al., 2013*). By incorporating various epitopes onto a homologous scaffold, the overall structural integrity of the antigenic surface can be maintained on the chimeric LigB Ig-like domain construct. Based on the high resolution LigB12 structure, each of the three chimeric segments encompassed a 900 to 2000 $\text{Å}^2$ surface and could potentially make a full conformational epitope (*Ptak et al., 2014*). While only two of three LigB chimeric segments were able to present as a full epitope experimentally, the third segment was required for binding by several of the mAbs. Indeed, initial experiments suggest that a redesigned chimera can harbor three separate epitopes on a single LigB Ig-like domain scaffold. Interestingly, the single domain LigB10-B7-B7 provides better protection to hamsters than two longer constructs, LigB7-12 (which provided poor protection) and LigB1-7 (which provided good protection) (*Yan et al., 2009*; *Conrad et al., 2017*). Exposure of epitopes in single domain antigens which would otherwise be blocked by host factors (*Beernink et al., 2011*) (i.e., ECM or serum proteins [*Lin et al., 2009a*; *Breda et al., 2015*; *Hsieh et al., 2016*]) in multi-domain constructs may enhance the efficiency. Two future goals of LigB recombinant vaccine studies will be to optimize further the chimeric antigens, consciously limiting the functionality of host-interacting sites, and to offer cross-species protection against different serovars.

This study and other recent studies (*Ptak et al., 2014*; *Conrad et al., 2017*) have significantly advanced our understanding of Lig protein structure and the potential of recombinant vaccines. The field of leptospirosis is poised to take advantage of the new insights and could make significant improvements in vaccines and other treatments to reduce the agricultural and human impact of the pathogen, *Leptospira*.

# Materials and methods

## Key resources table

| Reagent type (species) or resource | Designation | Source or reference | Identifiers | Additional information |
|---|---|---|---|---|
| gene (*Leptospira interrogans* serovar Pomona) | LigB (1-12) | GenBank | GenBank:FJ030916 | |
| recombinant DNA reagent | pET28-His-Sumo (plasmid) | other | | Re-engineered by Dr. Mao's lab (*Manford et al., 2010*) |
| strain, strain background (*Escherichia coli*) | *E. coli* Rosetta strain | Novagen | Novagen:70954 | |
| commercial assay or kit | Ni-NTA resin | Qiagen | | |
| peptide, recombinant protein | Sumo protease (Ulp-1) | other | | protein expression by Dr. Chang's lab utilizing pET28-His-Sumo vector provided by Dr. Mao's lab, and IPTG induction in Rosetta strain *E. coli* |
| commercial assay or kit | Superdex75 (size exclusion chromatography) | GE Healthcare | GE_Life_Sciences: 17517401 | |

*Continued on next page*

Continued

| Reagent type (species) or resource | Designation | Source or reference | Identifiers | Additional information |
|---|---|---|---|---|
| software, algorithm | BeStSel | DOI: 10.1073/pnas.1500851112 | | Uses CD spectra obtained from Aviv-201 spectropolarimeter |
| software, algorithm | Clustal Omega | DOI: 10.1093/nar/gkt376 | RRID:SCR_001591 | |
| software, algorithm | ATSAS Suite (GNOM; DAMMAVER; DAMMIF; AMBIMETER) | DOI: 10.1107/S1600576717007786 | RRID:SCR_015648 | |
| software, algorithm | RAW | DOI: 10.1107/S1600576717011438 | | |
| biological sample (*L. interrogans*) | LigB1-7 | this lab | | protein expression utilizing pET28-His-Sumo vector and IPTG induction in Rosetta strain *E. coli* |
| biological sample (*L. interrogans*) | LigB7-12 | this lab | | protein expression utilizing pET28-His-Sumo vector and IPTG induction in Rosetta strain *E. coli* |
| biological sample (*L. interrogans*) | LigB5-12 | this lab | | protein expression utilizing pET28-His-Sumo vector and IPTG induction in Rosetta strain *E. coli* |
| commercial assay or kit | Protein A/G Chromatography | ThermoFisher Scientific Pierce | | |
| cell line (*Mus musculus*) | hybridoma clones | William Davis Laboratory | | mAb-producing cell lines generated from BALB/c mice immunized with recombinant LigB antigen |
| antibody | HRP-conjugated anti-mouse IgG (goat polyclonal) | Invitrogen | RRID:AB_2533947 | (1:5000) |
| antibody | anti-LigB1-7 (mouse monoclonal) | this lab | | (1:500) |
| antibody | anti-LigB7-12 (mouse monoclonal) | this lab | | (1:500) |
| antibody | HRP-conjugated anti-hamster IgG goat polyclonal) | KPL | https://www.seracare.com/products/kpl-antibodies-and-conjugates/secondary-antibodies/anti-hamster-igg–h-l–antibody/ | (1:1000) |
| antibody | HRP-conjugated anti-hamster IgM goat polyclonal) | SouthernBiotech | https://www.southernbiotech.com/?catno=6060-05&type=Polyclonal | (1:1000) |
| antibody | FITC-conjugated anti-mouse IgG (goat polyclonal) | ThermoFisher Scientific | RRID:AB_2533946 | (1:1000) |
| antibody | isotypic control IgG (mouse Fc fragment) | ThermoFisher Scientific | RRID:AB_10959891 | |
| strain, strain background (*Leptospira biflex*) | *Leptospira biflex* | other | | Colony maintained in Dr. Chang's lab |
| strain, strain background (*L. interrogans*) | *Leptospira interrogans* serovar Pomona | this lab | | Colony maintained in Dr. Chang's lab |

*Continued on next page*

*Continued*

| Reagent type (species) or resource | Designation | Source or reference | Identifiers | Additional information |
|---|---|---|---|---|
| strain, strain background (*L. interrogans*) | *Leptospira interrogans* serovar Manilae M1307 | other | | Colony maintained in Dr. Gerald Murray and Dr. Ben Adler's labs, provided as a gift |
| biological sample (*Homo sapiens sapiens*) | Normal human serum | ImmunoReagents | ImmunoReagents: SP-001-VX10 | |
| biological sample (*Mesocricetus auratus*) | Golden Syrian Hamster | Harlan Sprague Dawley Laboratory | | 5 weeks old at initial subcutaneous vaccination |
| chemical compound, drug | Adjuvant 2% Alhydrogel | InvivoGen | | |

## Cloning, protein expression and purification

A series of single (LigB1 to LigB12), double (LigB1-2 to LigB11-12, excluding LigB5-6), five-domain (LigB1-5 to LigB8-12), and multiple (LigB1-7, LigB7-12, and LigB5-12) Ig-like domains of LigB from *Leptospira interrogans* serovar Pomona (GenBank, FJ030916) (illustrated in *Figure 1—figure supplement 1A*, *Figure 2—figure supplement 1A*, and *Figure 4—figure supplement 1*) were constructed on the vector pET28-His-Sumo between BamHI and HindIII (or XhoI) restriction enzyme sites as previously described (*Manford et al., 2010*; *Lin et al., 2009b*; *Lin et al., 2009a*). The constructed plasmids containing LigB genes were transformed to *E. coli* Rosetta strain and the protein expression was induced with 1 mM isopropyl β-D-thiogalactopyranoside (IPTG) at 20 °C for 16 hr. After the cells were lysed using a high-pressure cell disruptor, the cell lysates were spun down and the supernatants were purified by Ni-NTA resin (Qiagen). The His-Sumo tagged LigB proteins were eluted with phosphate buffered saline (PBS; 137 mM sodium chloride, 10 mM sodium phosphate, 2.7 mM potassium chloride, 1.8 mM potassium phosphate) containing 300 mM imidazole and then digested with Sumo protease Ulp-1 while dialyzing against PBS buffer at 4 °C overnight (*Manford et al., 2010*). Afterwards, the digested proteins were applied to a second Ni-NTA resin to remove the His-Sumo tag, and the untagged proteins were collected in the flow-through fraction. To gain higher protein purity, the untagged Lig proteins were further separated from other contaminates by size exclusion chromatography (SEC) Superdex75 (GE Healthcare), resulting in only one major species migrating on SDS-PAGE. For secondary structure analysis, circular dichroism (CD) spectra of LigB proteins were measured on an Aviv 202–01 spectropolarimeter (Aviv Biomedical, Lakewood, NJ) and predicted secondary structure composition was obtained using BeStSel (*Micsonai et al., 2015*). Chimeric LigB5/LigB12 and LigB7/LigB10 constructs were created by overlapping extension PCR to produce all six possible swapped genes from three protein segments (*Figure 6*). The three segments, strand A-C, strand C'-F and strand G-G', and specific residue boundaries were identified based on the high-resolution NMR structure of LigB12 (PDB ID 2MOG) (*Ptak et al., 2014*) and the sequence alignment of parent domains. The percent identity matrix for the set of Ig-like domains was obtained using the EMBL-EBI web service Clustal Omega (*McWilliam et al., 2013*). After ligation into pET28-Sumo vectors, the selected positive constructs were isolated from kanamycin resistant colonies and confirmed by DNA sequencing.

## SAXS construct window optimization

Constructs corresponding to two, five and eight domains were generated to identify the optimal domain number required to obtain useful SAXS-derived envelopes (*Figure 1—figure supplement 1*). The 8-domain long construct (LigB5-12) failed to generate envelopes with distinct domain regions representative of possible 8-domain structures. High average normalized spatial discrepancy (NSD) values for a reconstruction indicate increased variability among the trial models used to build the final envelope. Increasing average NSD with longer constructs may reflect a wider range of conformations expected with longer chains but additional factors could influence NSD statistics for longer constructs. The maximum diameter of a model is limited by the minimum q value measured in the scattering profile, as dictated by the Shannon Limit (*Putnam et al., 2007*). Thus, for these experiments, $q_{min} = 0.1$ implies that models can be no longer than $pi/0.1 = 314$ Å, under ideal conditions. In reality, minor aggregation and noise at low q may also place additional limits on model

length. Mylonas and Svergun have also shown that reconstructions of long rods are not as reliable as more compact structures (*Mylonas and Svergun, 2007*). For these reasons, we have chosen to assemble a full-length model from shorter overlapping constructs. SAXS envelopes derived from two or five sequential domains were generally comprised of distinct domain regions and accurately reflected the number and size of expressed domains. Of the 2- and 5-domain structures, only the envelopes of five sequential domains provided the relative orientation of multiple neighboring domains.

## Small angle X-ray scattering

LigB-derived proteins were exchanged into PBS buffer (pH 7.4) and concentrated to 12–20 mg/ml as assessed by absorption at $UV_{280}$. SAXS data for individual samples was measured for 15 scans at $1\times$, $2/3\times$, and $1/3\times$ protein concentrations after dilution with PBS buffer and centrifugation at 14,000 rpm for 10 min. Capillary cells were robotically loaded with 30 μL samples from a 96 well-plate maintained at 4°C (*Nielsen et al., 2012*). Between each sample, the capillary cell was thoroughly washed with detergent and water and then dried with air. All SAXS experiments were collected at the Cornell High-Energy Synchrotron Source (CHESS)'s F2 or G1 beamline using a dual Pilatus 100 K-S SAXS/WAXS detector (*Acerbo et al., 2015*; *Skou et al., 2014*). Background subtraction of SAXS buffer and further data and statistical analysis were performed using the free open-source software, RAW (*Nielsen et al., 2012*; *Hopkins et al., 2017*). The GNOM program from the ATSAS suite was used to determine P(r) plots. Optimal Rmax was determined by screening at 5 Å intervals and the P(r) plots were normalized to the first peak. Profiles best representing the dilute (ideal) solution limit were used to generate *ab initio* models with DAMMAVER and DAMMIF programs from the ATSAS suite (*Petoukhov et al., 2012*; *Volkov and Svergun, 2003*). 10 initial models were used to calculate the final model. Models with a normalized spatial distribution > mean + 2*standard deviation were treated as outliers and not included in determining the final model. No more than one model was excluded for each structure. Model uniqueness was evaluated using the AMBIMETER program (*Petoukhov and Svergun, 2015*). AMBIMETER score ranges < 1.5 indicate a unique ab initio shape determination, 1.5–2.5 indicate some potential for alternate solutions, and >2.5 indicate that multiple shape solutions will fit the data.

## Monoclonal antibody (mAbs) production

A 5 mg total of LigB1-7, and separately 5 mg of LigB7-12, was used to immunize and boost the antibody production from five BALB/c mice. Hybridoma clone generation was conducted in the laboratory of Dr. William Davis (fee-for-service) as previously described (*Park et al., 2015*). Standard ELISA assays were used for in-house screening of the hybridoma clones for positive supernatants. A total of 24 clones of IgG-type mAbs against LigB1-7 (library C; Lig protein Conserved region) and 36 clones of IgG-type mAbs against LigB7-12 (library V; Lig protein Variable region) were generated. Each library produced 9 mAbs which were found to be of moderate to high binding efficiency and utilized for further characterization. Purification of mAbs was conducted by protein A/G chromatography (Pierce) using the manufacturer-recommended procedure with minor modification (*Eliasson et al., 1989*). Briefly, the hybridoma supernatant was dialyzed against the binding buffer (100 mM potassium phosphate, 150 mM sodium chloride, pH 8.0) overnight at 4 °C. The dialyzed sample was applied to a protein A/G column (pre-equilibrated with binding buffer). After washing with 10–15 ml of binding buffer to remove the unbound fraction, the bound mAb was eluted with 100 mM glycine at pH 3.0 and neutralized immediately with 1 M Tris at pH 9.0. The eluate containing mAb was dialyzed against PBS buffer at pH 7.0, concentrated to 3–4 mg/ml, and stored at −80 °C.

## ELISA binding assay

LigB-binding mAbs were selected using an ELISA assay (*Lin et al., 2009a*; *Lin et al., 2011*). Microtiter plates (Nunc MaxiSorp, ThermoFisher Scientific) were coated with 1 μg of LigB1-7 or LigB7-12 in coating buffer (0.2 M NaHCO₃, pH 9.4) at 4°C overnight. After blocking with 3% BSA in PBS buffer for 1 hr, hybridoma supernatants were prepared at 1/500 dilutions and individually applied to LigB-coated wells for initial screening. To obtain dissociate constants, selected mAbs with moderate to high binding affinity to LigB were serially diluted in PBS (0.00686, 0.0137, 0.0412, 0.123, 0.370, 1.11, 3.33 and 10 μM) and then individually applied to LigB-coated wells. Between each step, PBS

containing 0.05% Tween-20 (PBS-T) was used to wash the plates for three times. Subsequently, anti-mouse IgG antibody conjugated with HRP (1:5000, Invitrogen) was added to detect the binding of mAbs presented in hybridoma supernatants. Finally, 100 µl of TMB peroxidase substrate (KPL) was applied to each well, the optical density of which was recorded at 630 nm by ELx808 Absorbance Microplate Reader (BioTek). $OD_{630}$ values represent the mean of three independent trials ± the standard deviation. For each trial, samples were assayed in two replicates.

For examining the domain specificities of mAbs, single domain LigB fragments (LigB1 to LigB12), double domain LigB fragments (LigB1-2 to LigB11-12, excluding LigB5-6), positive control (LigB1-7, LigB7-12), or negative control (BSA) were coated with a fixed concentration (1 µg/well) on the ELISA plates. Each mAb against LigB1-7 or LigB7-12 was then added to corresponding single-domain or double-domain LigB coated wells. For binding analysis of each chimera, the set of six chimeric LigB proteins plus the two parent LigB proteins were coated on the ELISA plates (1 µg/well). Each mAb with activity against either of the two parent LigB proteins was screened for binding activity on the LigB coated wells. For measuring the antibody responses triggered by LigB-based recombinant vaccines, sera from hamsters immunized with LigB7, LigB10 and LigB10-7-7 were collected at different time points (pre-immunization, post-immunization and post-booster). Then, these serum samples (1:500) were added to corresponding LigB Ig-like domain coated wells (1 µg/well), and also applied to LigB12 coated wells (negative control). Finally, anti-hamster IgG or IgM antibody conjugated with HRP (1:1000, KPL or SouthernBiotech) was used to detect the LigB-bound antibodies from hamster sera. All single and double domain ELISA experiments including single chimeric domain experiments were conducted in three trials, the mean ±S.D. of which were shown in bar charts.

## Surface binding of mAbs to live leptospira by flow cytometry

*Leptospira interrogans* freshly harvested from hamsters challenged with serovar Pomona or overnight NaCl-treated low passage *Leptospira* were prepared in PBS buffer containing 5 mM $MgCl_2$ at $10^8$ cells/ml. Various anti-LigB mAbs (100 µg/ml) were individually applied to the bacterial suspension at 1:500 dilution. After a 1 hr incubation at room temperature, the bacteria-mAbs mixtures were spun down at 2,000 g for 7 min and then washed with PBS containing 1% BSA. Subsequently, goat anti-mouse antibodies conjugated with fluorescein (FITC) (1:1000) were used as secondary antibodies for probing the bacteria-bound mAbs. After another PBS-BSA wash, the bacteria-mAbs mixtures were fixed by 0.5% formaldehyde in PBS. The non-pathogenic *Leptospira biflex*, which does not express LigB, was subjected to the same procedure by incubating with selected mAbs for the control experiments (*Figueira et al., 2011*). Another negative control was conducted by treating the pathogenic *Leptospira* with unrelated mouse IgG (isotypic mouse IgG purchased from ThermoFisher Scientific). Flow cytometry was performed at the Cornell University Flow Cytometry Core facility using a BD LSR-II (BD Biosciences) instrument with the excitation laser at 488 nm and the emission wavelength at 525/575 nm. Unstained *Leptospira* was identified by forward scatter (FSC) and side scatter (SSC). Selected mAb (C5) treated *L. interrogans* without secondary antibodies (FITC-conjugated goat anti-mouse IgG) was used as a negative control to set the gating region. The mean FITC-positive (MFI) count was obtained for the gated region of the 525 nm vs. 575 nm scatter plots using BD FACSDiva software. For each sample, at least 20,000 cells were analyzed in two independent trial of two replicates.

## Serum bactericidal assay

To examine if the mAbs were effective at killing bacteria in vitro, the serum bactericidal activity (SBA) assay was conducted. $10^8$ cells/ml of low passage, high virulent *L. interrogans* seorvar Pomona and Manilae strain M1307 were prepared in PBS buffer containing 2 mM $MgCl_2$ and 1 mM $CaCl_2$, and then mixed with respective mAbs, plus 25% of normal human serum (ImmunoReagents) as a complement source. The mixtures were incubated at 37°C for 1 hr, and the viability of the bacteria were examined using dark-field microscopy. The survival rate of leptospira was calculated as the number of motile (alive) cells in every 100 counts separately by two researchers (blind). The mean value was obtained from the two independent (blind) measurements as a single technical replicate. Three independent trials of two replicates were measured.

The viability of the leptospira was also accessed from the luminescence emitted by metabolically active *L. interrogans* Manilae strain M1307 (*Murray et al., 2010*). The same preparations were

examined at different time points (0, 30, 60 and 90 min), and the luminescence intensity of each sample was measured by GloMax 96 Microplate Luminometer (Promega). The survival rate of leptospira was calculated by the intensity at a specific time point divided by the intensity at 0 min. In addition, a series of dilutions of mAbs (100 µg/ml to 3.125 µg/ml) were evaluated for dose dependent bactericidal efficacy of antibodies. The time ($LT_{50}$) and dose ($LD_{50}$) required to reach 50% lethality were obtained from fitted logistic and dose inhibition curves, respectively (Origin 7.0).

### Hamster challenge

Hamsters (Harlan Sprague Dawley Laboratory) were housed in isolation units approved by the Cornell University Institutional Animal Care and Use Committee (Protocol number: 2015–0133). The Golden Syrian Hamsters used in vaccine trials were allowed to run free in the cage, were fed a commercial ration, and were provided water *ad libitum* as previously described (*Kunjantarachot, 2014*). Six 5-week-old hamsters each were vaccinated subcutaneously for each LigB-based recombinant vaccine containing adjuvant 2% Alhydrogel (InvivoGen) at 3 week intervals for a total of two injections. The control group was injected with adjuvant only. Three weeks after the final vaccination, all animals were challenged with $2.5 \times 10^2$ of triple passaged *L. interrogans* seorvar Pomona through the intraperitoneal route as previously described (*Kunjantarachot, 2014*). Kidneys, livers, lungs and urinary bladders were biopsied from hamsters within 1 hr after euthanasia (*Palaniappan et al., 2006*). Leptospiral loads found in livers, kidneys and urinary bladders from all immunization groups were examined by real-time RT-qPCR (*Levett et al., 2005*). The total RNA was extracted from target tissues and then reverse transcribed to cDNA. Subsequently, the *Leptospira* specific gene, LipL32 (*Haake et al., 2000*), was amplified and detected with fluorescence by 7500 Fast Real-Time PCR system. Histopathological tissue slices were fixed with 10% neutral buffered formalin and stained with hematoxylin and eosin. Tissue samples were imaged and scored by light microscopy.

## Acknowledgements

The authors thank the laboratory of Dr. William Davis for hybridoma generation, the Flow Cytometry Core at Cornell University for help with flow cytometry experiments, Dr. Cynthia Leifer and Jody Lopez for graciously providing access to their luminometer, and Dr. Holger Sondermann and Alex Devarajan for critically reading the manuscript. We also thank Dr. Gerald Murray and Dr. Ben Adler for generously giving luminescent *L. interrogans* Manilae strain M1307. CHESS is supported by the National Science Foundation (NSF) and NIH/NIGMS via NSF award No. DMR-1332208, and the MacCHESS resource is supported by NIGMS award GM-103485.

## Additional information

### Funding

| Funder | Grant reference number | Author |
| --- | --- | --- |
| Center for Advanced Technology (CAT) program | 478-3400 | Yung-Fu Chang |
| Biotechnology Research and Development Corporation | 478-9355 | Yung-Fu Chang |

The funders had no role in study design, data collection and interpretation, or the decision to submit the work for publication.

### Author contributions

Ching-Lin Hsieh, Conceptualization, Data curation, Formal analysis, Validation, Investigation, Methodology, Writing—original draft; Christopher P Ptak, Conceptualization, Data curation, Formal analysis, Validation, Investigation, Methodology, Writing—original draft, Writing—review and editing; Andrew Tseng, Igor Massahiro de Souza Suguiura, Tepyuda Sritrakul, Ting Li, Investigation; Sean P McDonough, Formal analysis, Investigation, Methodology; Yi-Pin Lin, Formal analysis, Writing—original draft; Richard E Gillilan, Formal analysis, Methodology; Robert E Oswald, Formal analysis, Validation, Writing—original draft, Writing—review and editing; Yung-Fu Chang, Formal analysis,

Supervision, Funding acquisition, Writing—original draft, Project administration, Writing—review and editing

## Author ORCIDs
Christopher P Ptak http://orcid.org/0000-0003-2752-0367
Richard E Gillilan http://orcid.org/0000-0002-7636-3188
Yung-Fu Chang http://orcid.org/0000-0001-8902-3089

## Ethics
Animal experimentation: Animals were housed in isolation units approved by the Cornell University Institutional Animal Care and Use Committee (Protocol number: 2015-0133).

## Decision letter and Author response
Decision letter https://doi.org/10.7554/eLife.30051.033
Author response https://doi.org/10.7554/eLife.30051.034

# Additional files

## Supplementary files
• Transparent reporting form
DOI: https://doi.org/10.7554/eLife.30051.027

## Major datasets
The following previously published datasets were used:

| Author(s) | Year | Dataset title | Dataset URL | Database, license, and accessibility information |
|---|---|---|---|---|
| Chang YF | 2008 | FJ030916: Leptospira interrogans serovar Pomona isolate pLPLIGB LigB (ligB) gene, partial cds. | http://www.ncbi.nlm.nih.gov/nuccore/FJ030916 | Publicly available at NCBI Nucleotide (accession no. FJ030916) |
| Ptak CP, Hsieh C, Lin Y, Maltsev AS, Raman, R, Sharma Y, Oswald RE, Chang Y | 2014 | Solution structure of the terminal Ig-like domain from Leptospira interrogans LigB. | http://www.rcsb.org/pdb/explore/explore.do?structureId=2MOG | Publicly available at the RCSB Protein Data Bank (accession no. 2MOG) |

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
