## [Decision Letter]

Thank you for submitting your article "Extended structure of a *Leptospira* antigen offers high bactericidal antibody accessibility amenable to vaccine design" for consideration by *eLife*. Your article has been reviewed by three peer reviewers, and the evaluation has been overseen by a Reviewing Editor and Wendy Garrett as the Senior Editor. The reviewers have opted to remain anonymous.

The reviewers have discussed the reviews with one another and the Reviewing Editor has drafted this decision to help you prepare a revised submission.

Leptospirosis is a widespread zoonotic bacterial infection that can cause serious disease in both animals and humans and can be refractory to treatment. Prevention is hampered by the fact that current vaccines are only partially effective. The work by Hsieh, et al., uses structural information of the immunoglobulin-like domains of a common *Leptospira* coat protein, LigB, to rationally design a recombinant multi-epitope antigen that appears to provide superior protection that either the full-length proteins or its individual domains. The present manuscript represents a dizzying amount of work, including comprehensive SAXS analysis of the multi-domain LigB protein and generation and characterization of two mAb libraries against LigB domains. Overall, the manuscript is well-written and easy to parse. Some questions, however, should be clarified.

Essential revisions:

1) While the authors make a point of highlighting the importance of the SAXS data to their design efforts, in reading the paper, it seems that the NMR structures of the individual domains were really the drivers (i.e., it really seems that the recombinant antigen could have been designed with only the NMR structure and the ELISA data). If the authors are presenting this as a roadmap to others, they might either better explain how the SAXS data really were critical, or change their language a bit.

2) In Figure 5, is it really appropriate to arbitrarily pick data in order to get a decent R^2^ for a linear fit? The authors briefly discuss the outlying mAbs – C12 and C23, but it seems that authors missed a huge opportunity in understanding what drives a much worse than expected and much better than expected activity vs. affinity, which has the potential to inform other rationally designed vaccination efforts.

3) While potentially outside of the scope of the present work, it seemed odd that the authors did not compare or even mention differences in memory response when using different vaccination strategies.

4) The main text does not state where the 60 antibodies are coming from. Why where there only 60? How many hybridomas have been screened in total? An explanation is only given in the Materials and methods. Additionally, the data for the screening of the antibodies should be presented in its entirety and not buried in the supplement. Only then the reader can fully appreciate how the lead antibodies have been identified. Perhaps a heatmap indicating the binding would be better suited to represent the data.

5) The positive control for bactericidal activity is a polyclonal hamster antibody, while the authors test mouse antibodies. Is it possible to discuss the lack of efficiency of the hamster polyclonal in greater detail? Why not use a mouse polyclonal as control?

6) When describing in Figure 5 and supplements the bactericidal activity of the antibodies the authors leave binding affinity completely out of the picture. Without the knowledge of affinity, the authors cannot be sure if the bactericidal activity is due the recognized epitope or affinity of individual antibodies.

7) In the vaccination experiment the authors describe that hamsters vaccinated with the chimera B10/7/7 have a better chance of survival when infected with *Leptospira* than hamsters vaccinated with B10 or B7, respectively. The better control would have been hamsters simultaneously vaccinated with a mixture of B10 and B7. Also, the ELISA in Figure 7—figure supplement 2 lacks controls. The authors should add to the LigB7 and LigB10 sera ELISA the same controls as for the chimera. Additionally, the result should be added to main figure in the text.

---

## [Author Response]

Essential revisions:1) While the authors make a point of highlighting the importance of the SAXS data to their design efforts, in reading the paper, it seems that the NMR structures of the individual domains were really the drivers (i.e., it really seems that the recombinant antigen could have been designed with only the NMR structure and the ELISA data). If the authors are presenting this as a roadmap to others, they might either better explain how the SAXS data really were critical, or change their language a bit.

We have adjusted our language regarding how the SAXS data was influential in developing the study. Specifically, we have changed the impact statement, the final sentence of the Abstract, and the final sentence of the Introduction. We agree that, in hindsight, the immunoreactivity mapping experiments could have been designed to inform the accessible features of the antigen without the SAXS data. Still, the initial SAXS study yielded a low-resolution structure that suggested generating monoclonal antibodies would be worth the expense for both LigB1-7 and LigB7-12 regions. This is now noted in the Discussion section. If the SAXS study had revealed that some domains were clearly buried, then these regions could have been excluded from the generation of the mAb library. The SAXS data and immunoreactivity data offer complementary views of the overall structural architecture which also creates elements of redundancy.

2) In Figure 5, is it really appropriate to arbitrarily pick data in order to get a decent R^2^ for a linear fit? The authors briefly discuss the outlying mAbs – C12 and C23, but it seems that authors missed a huge opportunity in understanding what drives a much worse than expected and much better than expected activity vs. affinity, which has the potential to inform other rationally designed vaccination efforts.

For Figure 5, we agree with the reviewers that outliers should not be picked arbitrarily and also agree that identified outliers could be useful and informative. We now use a statistical method to determine the outliers in both Figure 5 correlation plots. A supplemental figure (Figure 5—figure supplement 2) has been added to show the Cook’s distance for the data in the correlation plots. For Figure 5 in particular, the outliers greatly influence the regression line so the R^2^ is reported for both the total dataset and the dataset without the statistically-determined outliers.

Additionally, we have included further discussion with regard to the identified outliers, C12 and C23. Please note that we have changed the Y-axis of the plot in Figure 5 from a relative ELISA binding measurement to a K_D_ based on ELISA. While C12 was determined to be a statistical outlier in both the original and new Figure 5 plots, the K_D_ value of C12 is closer to the expected value from the fitted correlation. We note that C12 binds to two double and two single LigB Ig-like domains within LigB1-7 (not LigB7) and therefore may bind to a conserved region of the domains. To test if C12 was promiscuous enough to also bind to LigB7-12, an additional ELISA was performed to assess the possibility of interactions between library C mAbs and LigB7-12. C12 was the only mAb with significant binding to LigB7-12 suggesting that the ability to bind to several LigB Ig-like domains allows C12 to more effectively kill bacteria. A supplemental figure (Figure 5—figure supplement 3) has been added to show C12 binding to LigB7-12.

The other statistically identified outlier, C23, is less effective in bactericidal activity than expected based on both antigen binding and cell surface binding experiments. We suggest that the kinked structure of the LigB domains at the C23-binding site may play a role in decreasing its ability to initiate the cell death inducing cascade. The steric restriction combined with the target domain’s proximity to the leptospiral surface could slow mAb accessibility. Assays for antigen and cell surface binding are less dependent on time as incubation times were long enough to reach equilibrium. The bactericidal activity assay was inherently dependent on time which is compounded by time delays imposed by activation of the downstream complement cascade. We analyzed the correlation between LD_50_ vs. LT_50_ in the bactericidal assay using Cook’s distance and found C23 to be the only statistical outlier (Figure 5—figure supplement 4). In a plot of K_D_ vs. LT_50_ (Figure 5—figure supplement 4), C23 displays even weaker bactericidal activity relative to other mAbs than in the K_D_ vs. LD_50_ plot. The effect of time on the bactericidal activity of C23 is larger than for other mAbs. One possible reason, the LD_50_ measurement was assessed at the longer time point of 90 minutes while LT_50_ is also influenced by measurements at 60 minutes. The comparison of bactericidal activity measurements suggests that the bactericidal activity of C23 could be impaired by conformationally inhibited binding leading to slower than expected bacterial killing.

3) While potentially outside of the scope of the present work, it seemed odd that the authors did not compare or even mention differences in memory response when using different vaccination strategies.

We include a figure to illustrate the relative IgG and IgM response to LigB10-B7-B7 immunization (Figure 7—figure supplement 3). As expected, a larger secondary immune response is observed for IgG only. The additional data is now mentioned in the Results section titled “Chimera LigB10-B7-B7 confers enhanced protection against *Leptospira* lethal challenge”.

4) The main text does not state where the 60 antibodies are coming from. Why where there only 60? How many hybridomas have been screened in total? An explanation is only given in the Materials and methods. Additionally, the data for the screening of the antibodies should be presented in its entirety and not buried in the supplement. Only then the reader can fully appreciate how the lead antibodies have been identified. Perhaps a heatmap indicating the binding would be better suited to represent the data.

We have added a Results section specifically for the description of the mAb library creation and screening. Figure 2 (previously a supplement) has become a main text figure and subsequent figures have been renumbered accordingly. Figure 2 depicts the ELISA screen and cutoff used to select hybridomas for detailed analysis. For each library, two sets of 12 hybridoma supernatants were tested in the initial screen. An additional 12 hybridomas were needed to be screened in library V (LigB7-12) to identify 9 total mAbs with the desired ELISA cutoff (OD_630_ = 1). The screening data is presented in its entirety. We now provide more detail on how the antibodies were obtained in both the added Results section and the Materials and methods. A heatmap of the data was unable to depict the ELISA cutoffs so we are presenting the data as a bar graph. An additional supplementary figure (Figure 2—figure supplement 1) was added to show the distribution of relative binding efficiencies. This histogram helps to illustrate how the lead mAbs were identified.

5) The positive control for bactericidal activity is a polyclonal hamster antibody, while the authors test mouse antibodies. Is it possible to discuss the lack of efficiency of the hamster polyclonal in greater detail? Why not use a mouse polyclonal as control?

In the serum bactericidal assay, the no antibody and isotypic mouse IgG groups are the true controls for the assay. We now do not refer to the hamster anti-LigB polyclonal antibodies as a control in the manuscript. We agree with the reviewers that the mouse pAb would have been a better control; Unfortunately, mouse pAbs were not retained during the mAb library creation and are therefore not readily available. Hamster anti-LigB pAb were retained from vaccine trials and was easily included in the bactericidal assay. Because the result suggests that the mAbs generated in this study are more effective at killing *Leptospira* than hamster anti-LigB pAb that would be generated using the conventional leptospirosis animal model, we felt the data should be included but we are willing to remove the data if the reviewers would prefer taking it out. There are three possible explanations for the lack of hamster pAb efficiency. First, the pAbs are likely to include both high affinity and low affinity anti-LigB antibodies and also background antibodies with no affinity for LigB. The effectiveness of pAbs are likely to be reduced because the concentration of high affinity anti-LigB antibodies is lower for a pAb solution of the same overall concentration as for a solution the purified mAbs. Second, pAbs contain a mixture of IgG subclasses (IgG1, IgG2a, IgG2b and IgG3). The ability of the different hamster-derived IgG subclasses to effectively activate the complement system is unknown; however, different IgG subclasses from both mouse and human exhibit variability in their effectiveness at activating the complement system. Third, while the ability of mouse antibodies to activate the human complement cascade is known, the relative ability of hamster antibodies to activate the same cascade has not been described. Because we do not know if mouse and hamster antibodies are equally efficient at eliciting the killing response, we cannot rule out the possibility that hamster antibodies are less effective in the assay. We now discuss the possible reasons for the lack of hamster pAb efficiency in the Results section of the manuscript (titled “Bactericidal activity of anti-LigB mAbs”).

6) When describing in Figure 5 and supplements the bactericidal activity of the antibodies the authors leave binding affinity completely out of the picture. Without the knowledge of affinity, the authors cannot be sure if the bactericidal activity is due the recognized epitope or affinity of individual antibodies.

We agree with the reviewers that binding affinity (K_D_) is a more accurate measurement of mAb-antigen interactions than the ELISA-derived relative antigen binding measurement. We have replaced the relative binding measurements with K_D_ values. K_D_ values for purified mAbs were determined by fitting concentration-dependent ELISA measurements. We have replaced the relative binding measurements with the concentration-dependent ELISA curves and K_D_ values (new Figure 3 and Figure 3—figure supplement 1). K_D_ values have also replaced the relative binding measurements in Table 1 and the correlation plot with LD_50_ for bactericidal activity (Figure 5). Overall, the LD_50_ correlation plot using K_D_ is similar to the plot using relative ELISA binding measurements with C12 and C23 being identified as the only two statistical outliers in both plots. The K_D_ vs. LD_50_ shows a high enough correlation (R^2^ = 0.773) to suggest bactericidal activity is dependent on binding affinity for most mAbs. The outliers, C12 and C23, were explored further in the three added supplements to Figure 5 to understand the basis for their non-conformity.

7) In the vaccination experiment the authors describe that hamsters vaccinated with the chimera B10/7/7 have a better chance of survival when infected with Leptospira than hamsters vaccinated with B10 or B7, respectively. The better control would have been hamsters simultaneously vaccinated with a mixture of B10 and B7. Also, the ELISA in Figure 7—figure supplement 2 lacks controls. The authors should add to the LigB7 and LigB10 sera ELISA the same controls as for the chimera. Additionally, the result should be added to main figure in the text.

We have added the requested controls for Anti-LigB7 and Anti-LigB10 sera ELISA to Figure 7—figure supplement 2. A reference to the figure supplement was added to the main figure legend.

There are several reasons why we feel that the benefits of the additional control would not be an ideal control and do not warrant significantly delaying the publication of this manuscript. From a manufacturing standpoint, purification and subsequent mixing of domains would be costlier than purification of a single domain. This control would not generate a useful result in and of itself. Additionally, we show a robust gain in vaccine efficiency and the importance of a positive control would be minimal. Also, since all of the vaccines used in the trial were single domain proteins from a single purification, the inclusion of a mixed protein solution would not be a direct comparison. While a vaccine trial that included another control with a mixture of LigB7 and LigB10 could have been added, the additional vaccine trial would require 3-4 months to complete. Again, we argue that the small benefit of the additional vaccine trial control does not warrant the time commitment and delay that it would require.